# Adult neurogenesis reconciles flexibility and stability of olfactory perceptual memory

**Bennet Sakelaris, Hermann Riecke***

Engineering Sciences and Applied Mathematics, Northwestern University, Evanston, United States

## eLife Assessment

In this **important** study, the authors use computational modeling to explore how fast learning can be reconciled with the accumulation of stable memories in the olfactory bulb, where adult neurogenesis is prominent. Their model demonstrates that changes in excitability, plasticity, and susceptibility to apoptosis during the maturation of adult-born granule cells can help resolve the flexibility-stability dilemma. These **compelling** results provide a coherent picture of a neurogenesis-dependent learning process that is consistent with diverse experimental observations and may serve as a foundation for further experimental and computational studies.

***For correspondence:**
h-riecke@northwestern.edu

**Competing interest:** The authors declare that no competing interests exist.

**Abstract** In brain regions featuring ongoing plasticity, the task of quickly encoding new information without overwriting old memories presents a significant challenge. In the rodent olfactory bulb, which is renowned for substantial structural plasticity driven by adult neurogenesis and persistent turnover of dendritic spines, we show that by synergistically combining both types of plasticity, this flexibility-stability dilemma can be overcome. To do so, we develop a computational model for structural plasticity in the olfactory bulb and show that it is the maturation process of adult-born neurons that enables the bulb to learn quickly and forget slowly. Particularly important are the transient enhancement of the plasticity, excitability, and susceptibility to apoptosis that characterizes young neurons. The model captures many experimental observations and makes a number of testable predictions. Overall, it identifies memory consolidation as an important role of adult neurogenesis in olfaction and exemplifies how the brain can maintain stable memories despite ongoing extensive neurogenesis and synaptic plasticity.

## Introduction

The learning and subsequent retention of information are fundamental tasks of the brain. Thus, plasticity during learning must allow for the rapid acquisition of new memories without quickly overwriting existing ones. When memories differ in valence, for example by an associated reward, more important memories can be specifically protected against overwriting by less important new information. Often, however, an animal is continually presented with new information without such distinguishing cues. In these situations, the brain faces the 'flexibility-stability dilemma' (**Grossberg, 1982**): it has to flexibly encode new memories while preserving the stability of previous ones.

The flexibility-stability dilemma can classically be resolved through the process of systems consolidation where memories are rapidly encoded by the hippocampus and gradually transferred to the neocortex where they remain quite stable (**McClelland et al., 1995**; **Roxin and Fusi, 2013**). In systems where consolidation is restricted to the very circuit that encoded the information in the first place, it

is not yet well understood how this issue is combated. It has previously been addressed in models built on theoretical, complex synapses that are characterized by cascades of depressed and potentiated states (*Fusi et al., 2005*; *Benna and Fusi, 2016*); however, detailed experimental evidence for systems in which this local mechanism resolves this issue without making use of systems consolidation is not yet available.

Here, we address this issue by considering perceptual learning in the olfactory system. Perceptual learning is a form of continual learning in which an animal learns to discriminate between similar stimuli through repeated exposure to the stimuli without the stimuli being associated with any explicit rewards or aversions that may modulate plasticity and thus prioritize or protect specific memories. In the olfactory system, certain types of perceptual learning have been shown to occur in the olfactory bulb (OB; *McNamara et al., 2008*; *Gschwend et al., 2015*), which is renowned for its high degree of structural plasticity, most notably through adult neurogenesis. Experiments have demonstrated that adult neurogenesis is necessary for this learning (*Moreno et al., 2009*; *Li et al., 2018*), and computational modeling provides some understanding of the underlying mechanisms (*Chow et al., 2012*; *Adams et al., 2019*; *Shani-Narkiss et al., 2020*; *Kersen et al., 2022*). However, these behavioral observations could also be explained by synaptic plasticity alone (*Sailor et al., 2016*; *Meng and Riecke, 2022*). What, then, is the purpose of adding large numbers of new neurons? And why remove a sizable fraction of them again later?

Using a computational model of the OB, we demonstrate that the synergistic interaction between adult neurogenesis and synaptic plasticity can provide a massive computational advantage in olfactory memory by ameliorating the flexibility-stability dilemma. Importantly, it is not the adding of neurons as such but the maturation process of the adult-born neurons that achieves this goal: the experimentally observed heightened excitability and plasticity of newly arrived young cells (*Kelsch et al., 2009*; *Nissant et al., 2009*; *Wallace et al., 2017*) allow them to rapidly store new memories when they are young; the memories then stabilize as the aging neurons become less plastic. Furthermore, the transiently increased excitability of new neurons is important for helping them integrate into the network, while on longer time scales, their higher rate of apoptosis is needed to remove unnecessary neurons that would interfere with the ability of newer neurons to integrate when learning new odors.

Our biophysically motivated model captures a host of recent experimental observations, including how adult-born neurons are preferentially recruited to process new odors (*Moreno et al., 2009*; *Forest et al., 2020*; *Forest et al., 2019*), and how the silencing of these neurons extinguishes memory (*Forest et al., 2020*). Additionally, young neurons are highly sensitive to retrograde interference and die if a new odor is presented without the previously presented odor still being present in the environment (*Forest et al., 2019*). Moreover, our model makes multiple experimentally testable predictions, including that the rapid re-learning of a forgotten odor pair (*Sultan et al., 2010*) is enabled by the sensory-dependent dendritic elaboration of neurons that initially encoded the odors, and that the observed rapid re-learning would occur even if neurogenesis were blocked following the first enrichment and even though the initial learning did require neurogenesis. Furthermore, long time periods without odor enrichment or without apoptosis are predicted to negatively impact the subsequent ability to learn new odors.

## Results

We studied the effects of adult neurogenesis in a structurally constrained computational model that was designed to include many important, specific biological aspects of the olfactory bulb. Previous theoretical work has found conflicting results regarding the effects of new neurons on old memories, and the findings seem to depend sensitively on the implementation of neurogenesis, network architecture, and task (*Aimone and Gage, 2011b*). Although simpler models could be used to investigate adult neurogenesis, to gain biological insight, it is therefore essential to study neurogenesis in a model derived from the relevant biological processes and tasks.

### Formulation of the model

We considered an OB network comprising two distinct layers of neurons: the primary layer consisting of excitatory mitral cells (MCs) and the secondary layer composed of inhibitory granule cells (GCs), the preeminent neurogenic population in the OB. We omit preprocessing in the glomerular layer,

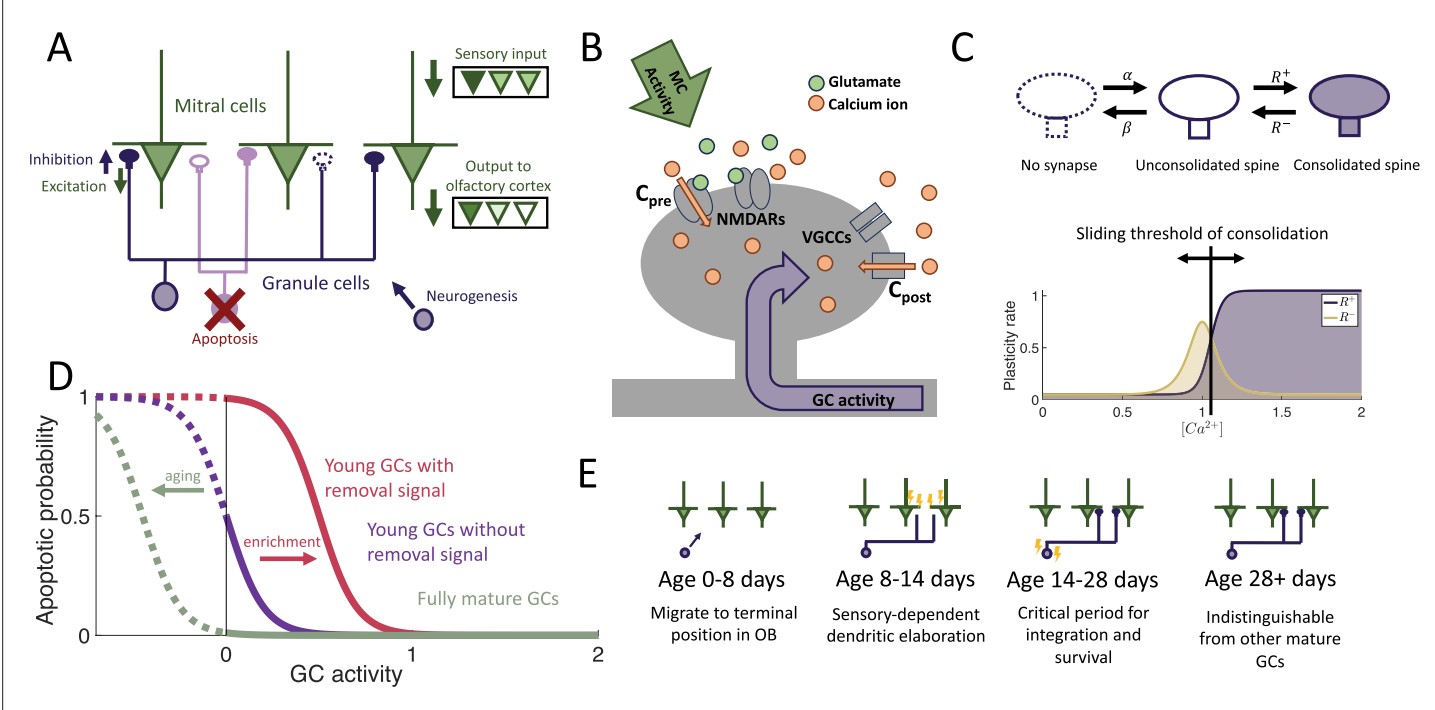

**Figure 1.** Computational model. (**A**) MCs relay stimuli to cortex. Reciprocal synapses with GCs can be functional or non-functional (**C**). Adult neurogenesis adds GCs and apoptosis removes GCs. (**B**) Calcium controls synaptic plasticity (***Graupner and Brunel, 2012***). Influx into spine through MC-driven NMDARs and through voltage-gated calcium channels (VGCC) opened by global depolarization of GCs. (**C**) Unconsolidated spines are formed with rate $\alpha$ and removed with rate $\beta$. Spines become consolidated with a rate $R^+$ and deconsolidated with rate $R^-$ (Top). $R^\pm$ depend on the local calcium concentration in the spine (Bottom). (**D**) GCs are removed with a rate that depends on activity and age of the cells, as well as environmental factors (see Materials and methods). (**E**) Development of abGCs. At age 8–14 days, they integrate silently into the OB. The formation and elaboration of their dendrites depends on sensory input. During their critical period (14–28 days), the abGCs are more excitable and plastic and have a higher rate of apoptosis. Beyond 28 days the abGCs are mature GCs.

although it also involves adult-neurogenesis, but on a much smaller scale (***Lledo et al., 2006***). In each timestep, new adult-born granule cells (abGCs) were added to the GC layer. The MCs received sensory input via glomerular activation patterns, forming representations of the stimuli in terms of the MC activity. These representations were shaped by inhibition from GCs, mediated through fully reciprocal synapses—each connection from a MC to a GC was paired with a reciprocal connection from that GC back onto the same MC. This bidirectional connectivity allowed GCs to both integrate and modulate MC activity, thereby reformatting sensory representations before they were projected to the olfactory cortex for further processing (***Figure 1A***).

The sparse structural characteristics of the dendritic networks were incorporated within both the MC and GC populations by allowing each GC to form synaptic connections with only a subset of MCs, which we refer to as the dendritic field of the GC. The MC-GC synaptic network was persistently modified by activity-dependent structural synaptic plasticity in which synapses located on GC dendrites were formed and eliminated (***Livneh and Mizrahi, 2012***; ***Sailor et al., 2016***). Because such structural plasticity is known to rely on the local calcium concentration at the spine (***Kasai et al., 2021***), we adapted a previously published model that approximates the calcium concentration at each synapse as a function of pre- and post-synaptic activity (***Graupner and Brunel, 2012***; ***Figure 1B***) and Synaptic plasticity in Materials and methods.

The synaptic dynamics were modeled with a Markov chain consisting of three states: non-existent, unconsolidated, and consolidated (***Figure 1C***). Non-existent synapses represent locations where a synapse between an MC and a GC is geometrically possible but not realized. Unconsolidated synapses represent filopodia, silent synapses, or unconsolidated spines that provide a foundation for a functional synapse but do not yet generate a postsynaptic response. Finally, consolidated synapses represent fully functional connections. We assumed that synapses transition between the unconsolidated

and non-existent states at a constant rate, independent of activity. Conversely, transitions between unconsolidated and consolidated synapses occurred with an activity-dependent rate so that sensory experience shapes the functional connectivity of the network. This assumption arises from experimental findings that GC spine dynamics depend on GC activity (*Breton-Provencher et al., 2016*; *Breton-Provencher et al., 2014*; *Saha et al., 2021*).

In the OB, apoptosis is activity-dependent (*Benson et al., 1984*; *Yamaguchi and Mori, 2005*; *Lin et al., 2010*; *Yokoyama et al., 2011*) with young abGCs more susceptible than mature GCs (*Yamaguchi and Mori, 2005*). Furthermore, the survival of abGCs has been shown to be modulated by behavioral state (*Yokoyama et al., 2011*; *Yamaguchi et al., 2013*; *Komano-Inoue et al., 2014*) and impacted by enrichment with novel, but not with familiar odors (*Veyrac et al., 2009*). Therefore, as a minimal model of apoptosis, we assumed that fully mature GCs had a low threshold of activity required for survival, while young abGCs had a higher threshold. In addition, the latter were susceptible to an apoptotic signal that was triggered by olfactory enrichment, which further raised the threshold, similar to the two-stage model for GC elimination proposed by *Yokoyama et al., 2011*; *Yamaguchi et al., 2013*; *Komano-Inoue et al., 2014*; *Figure 1D*.

To investigate the role of the aforementioned transient properties of newly arrived abGCs (summarized in *Figure 1E*), we evaluated the network's performance in a perceptual learning task where the network learns to discriminate between two similar stimuli purely through brief, repeated exposure to these stimuli (*Figure 2B*). In the olfactory bulb, it has been established that adult neurogenesis is required for such a task (*Moreno et al., 2009*), but not if there is reward (*Imayoshi et al., 2008*). This suggests that a major function of neurogenesis lies not in reinforcement-based behavior, but in refining sensory representations through mere experience. To simulate this implicit learning, the network is repeatedly presented with brief, alternating, similar artificial stimuli ('enrichment') and assessed in its ability to discriminate between these stimuli by comparing their MC-representations. We quantified this discriminability in terms of the Fisher discriminant (Appendix 1). The enhancement in discriminability parallels the formation of an odor-specific network structure. We therefore quantified the memory in terms of the specificity of that connectivity (Memory in Materials and methods).

## Age-dependent plasticity and excitability reconcile flexibility and stability of memory

In order to establish what can be gained from the addition of new neurons alone, we initially disregarded apoptosis and the transient properties of abGCs, resulting in a baseline model of adult neurogenesis where a homogeneous population of GCs grew over time. We define a neurogenic network as one in which the number of GCs grows linearly over time, and a non-neurogenic network as one with a fixed GC population size corresponding to the size of the neurogenic network at initialization. In *Figure 2C*, we simulated the perceptual learning experiment using neurogenic (solid lines) or non-neurogenic (dashed lines) networks comprising either uniformly fast (green) or slow (blue) synapses. There was no significant difference between the results of the neurogenic and non-neurogenic models, so neurogenesis alone did not impact learning or memory.

Due to the pivotal roles of plasticity rates in memory encoding, we first explored the impact of the differing rates of synaptic plasticity in young and old GCs (*Figure 2A*). The rates of spine turnover were calibrated to align with empirical data (*Sailor et al., 2016*; *Figure 2—figure supplement 1A*). This transiently enhanced plasticity partially resolved the flexibility-stability dilemma, as the age-dependent network encoded memories more swiftly than the network of slow synapses while maintaining greater stability than the network with fast synapses (*Figure 2C*, red curves). Thus, the heightened plasticity of young abGCs allowed them to rapidly encode new memories and retain them as the neurons matured and their plasticity rate decreased.

In addition to the stronger plasticity, abGCs exhibit increased excitability during their critical period. Mechanistically, excitability and plasticity are distinct: excitability governs a neuron's likelihood of firing in response to input, whereas plasticity determines the rate and extent of synaptic change given that activation occurs. To assess the impact of the increased excitability, we repeated the simulations incorporating it in addition to the enhanced plasticity (*Figure 2D*). Now, the model achieved a higher initial memory strength, which was on par with that reached by the fast network, while retaining the stability of the slow network (*Figure 2E*, red curve). Importantly, this only occurred in conjunction with age-dependent plasticity and not in the networks with constant plasticity rates (*Figure 2E*, blue

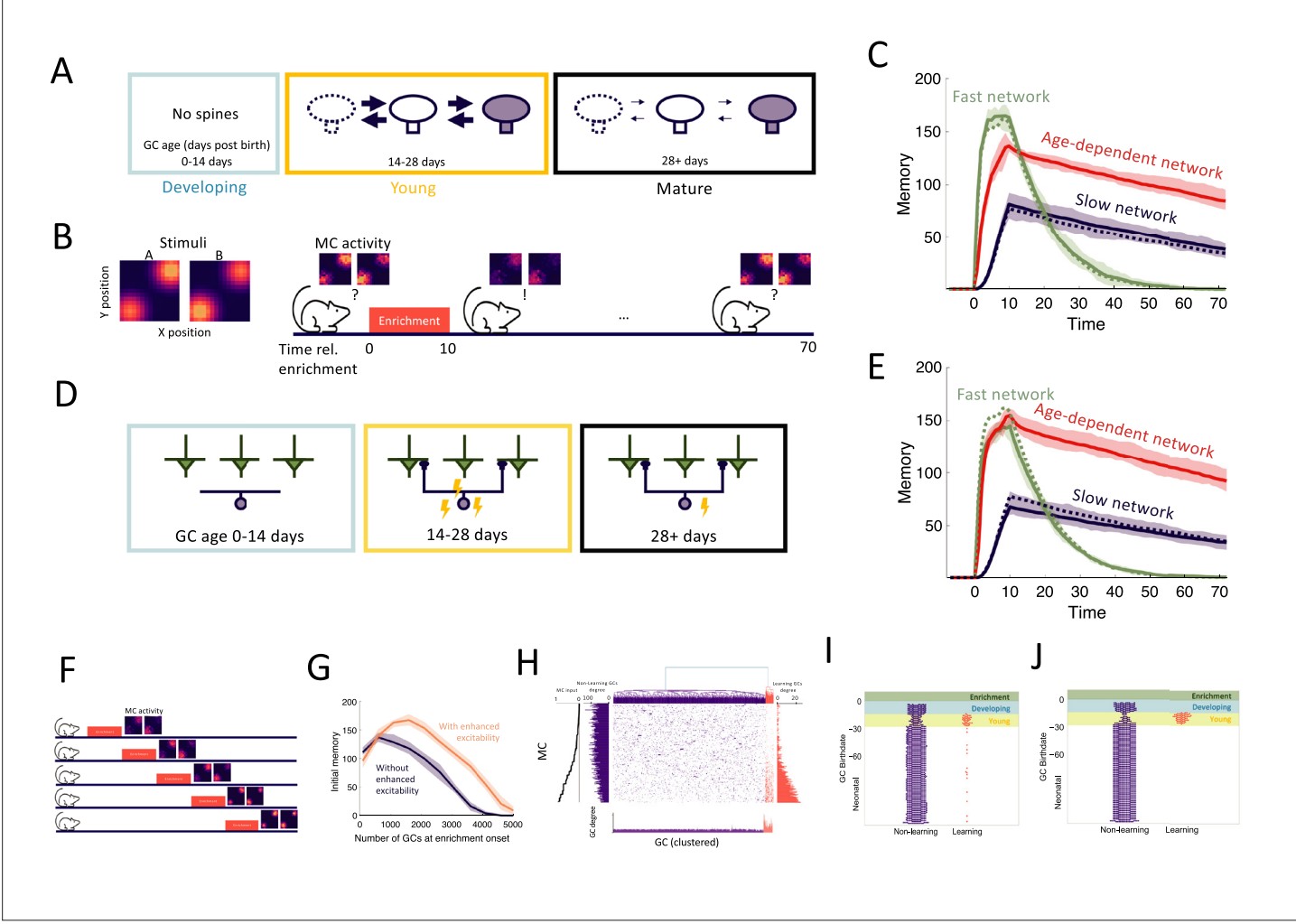

**Figure 2.** Age-dependent plasticity. (**A**) AbGCs in their critical period (14–28 days) exhibit greater spine turnover. (**B**) Left: Activity of MCs arranged on a two-dimensional grid and stimulated with stimulus A or B, respectively. Each pixel represents a single MC, and the arrangement is for visualization only. Right: MCs initially respond similarly to both stimuli, but responses diverge after a 10 day enrichment. Over time, spontaneous synaptic changes lead to forgetting. (**C**) Memory is measured as a function of the network connectivity (see Materials and methods) for three different models: neurons with fast plasticity (green), neurons with slow plasticity (blue), neurons with age-dependent plasticity (red). Dashed curves are networks without neurogenesis. Lines: mean across eight trials, shaded areas: full range. The memory evolution is similar to that of the odor discriminability as measured using the Fisher discriminant (*Figure 2—figure supplement 1B*). (**D**) abGCs also exhibit increased excitability during their critical period. (**E**) The initial memory is enhanced by the increased excitability. Lines: mean across eight trials, shaded areas: full range (**F**) Delayed enrichment simulation in which the model was allowed to grow for longer and longer times preceding enrichment. MC responses to the test odors show diminished learning for delayed enrichment. Lines: mean memory across eight trials, shaded areas: full range. (**G**) The memory immediately following enrichment as a function of the number of GCs at enrichment onset for the case with (orange) and without (purple) age-dependent excitability. Lines: mean across eight trials, shaded areas: full range. (**H**) Reciprocal MC-GC connectivity at the end of the enrichment period of one simulation; orange: odor-specific connectivity ('learning' GCs), purple: unspecific connectivity ('non-learning'). Center: connectivity matrix. Top: dendrogram reflecting hierarchical clustering of GCs according to their connectivity. Bottom: number of connections of each GC. Sides: number of connections of each MC to learning and non-learning GCs, respectively. MCs sorted according to input strength (leftmost panel). (**I,J**) Birthdates relative to enrichment onset of learning (orange) and non-learning (purple) GCs for models without and with increased excitability, respectively. At enrichment onset, GCs in the blue and yellow regions were in the developing and young stages, respectively (**A**). Green region shows enrichment period. Only GCs that were incorporated into the OB network (age >14 days) at the end of enrichment are shown. $N_{conn} = 100$ and $R_0 = 0.005$ were used for all simulations in this figure.

The online version of this article includes the following figure supplement(s) for figure 2:

**Figure supplement 1.** Spine turnover and consistency of memory measure.

**Figure supplement 2.** Buildup of abGCs interferes with learning.

and green curves). Thus, the increased excitability in young abGCs cooperates with their increased plasticity to improve the initial memory of the OB.

Models of adult neurogenesis often have a common drawback: as the networks grow, their performance tends to decrease due to greater interference by the accumulating adult-born neurons (*Meltzer et al., 2005*; *Aimone et al., 2011a*; *Kudithipudi et al., 2022*). We hypothesized that the increased excitability of young abGCs may mitigate this effect by raising the activity of abGCs that would otherwise be quieted through lateral inhibition from the vast population of mature GCs. To test this, we conducted a simulation where the onset of enrichment was progressively delayed, allowing an increasing number of GCs to accumulate in the OB (*Figure 2F*). We carried this out in cases with and without enhanced excitability of young abGCs. In keeping with previous results, accumulation of adult-born neurons eventually prevents the network from learning (*Figure 2G*). Here, this is due to excessive inhibition that suppresses the activity of new abGCs, preventing them from becoming selectively tuned to stimuli (*Figure 2—figure supplement 2*). However, the network featuring age-dependent excitability substantially outperformed the network without this feature. Thus, the transiently enhanced excitability of abGCs contributed to maintaining the learning flexibility of the OB in the face of persistent neurogenesis.

To better understand how memories were encoded in the OB, we analyzed the MC-GC connectivity immediately following enrichment. We performed hierarchical clustering on the columns of the connectivity matrix, in order to cluster GCs into groups of similar connectivity. There were two distinct clusters: a large portion of GCs were non-specifically connected (purple in *Figure 2H*), while others exhibited an increased total number of synapses and were preferentially connected with MCs that responded strongly to the enrichment odors (orange in *Figure 2H*). Thus, due to the reciprocal nature of all synapses, the mutual disynaptic inhibition was enhanced among the odor-responsive MCs. Cluster membership was then sorted by GC birthdate relative to enrichment, which revealed

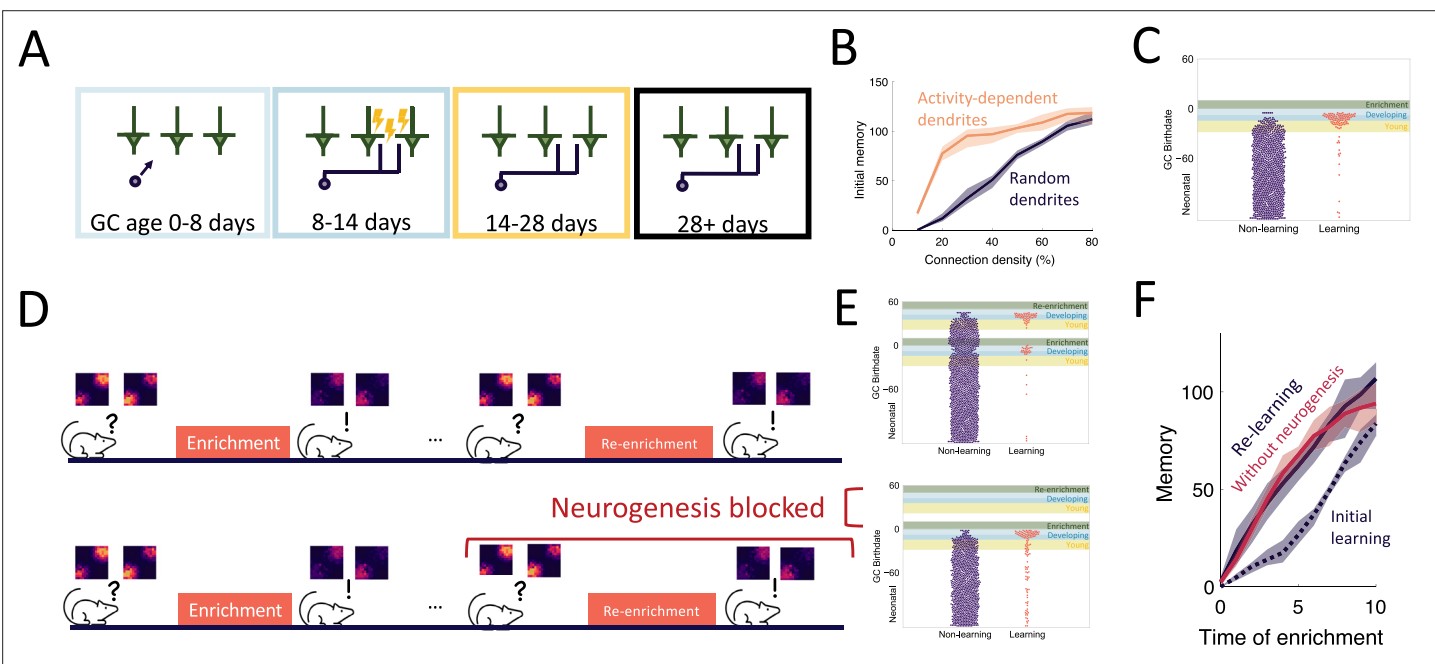

**Figure 3.** Dendritic structure. (**A**) Sensory-dependent silent integration of juvenile abGCs (*Figure 2A and D* and Materials and methods). (**B**) Memory following enrichment as a function of the number of potential synapses with random (purple) and activity-dependent (orange) dendritic elaboration. Lines: mean across eight trials, shaded areas: full range. (**C**) Birthdate analysis as in *Figure 2I*. Learning mostly by abGCs that develop their dendrites during enrichment (blue region). (**D**) Enrichment was followed by a period of spontaneous activity until the memory cleared, then re-enrichment occurred with the same set of stimuli. (**E**) Birthdate analysis with a second set of colored regions corresponding to the re-enrichment (**C**). (**F**) Memory evolution during the initial enrichment (dotted line) and second enrichment both with (purple solid line) and without (orange solid line) neurogenesis. Lines: mean across eight trials, shaded areas: full range.

The online version of this article includes the following figure supplement(s) for figure 3:

**Figure supplement 1.** Dependence of memory on dendritic development.

that abGCs that were young at the beginning of the enrichment period (yellow shaded area) were preferentially recruited to the learning cluster (*Figure 2I*). While the degree of overall recruitment was similar, the degree of preferential recruitment was increased through the transient hyperexcitability (*Figure 2J*). Thus, age-dependent excitability leads to the greater preferential recruitment of young, rapidly learning adult-born neurons, which in turn strengthens initial memories. Separately, as a result of this learning, synapses of odor-responding MCs were redistributed from the non-learning cluster to the learning cluster (*Figure 2H*, vertical MC degree bar plots). This is similar to findings in the hippocampus that young adult-born neurons both add new synapses to the network and replace existing synapses formed by mature GCs (*Adlaf et al., 2017*).

Notably, even with increased initial excitability, abGCs that arrived in the OB during the enrichment phase (blue area) made only a minimal contribution to the memory (*Figure 2I and J*). This conflicted explicitly with the experimental finding that the silencing of these abGCs extinguishes this memory (*Forest et al., 2020*; *Forest et al., 2019*). We therefore asked whether this discrepancy could be reconciled by considering additional known developmental properties of abGCs.

## The dendritic structure of abGCs latently encodes memories

An important aspect of GC development is the period of dendritic development when newborn neurons are integrating silently into the OB. Because of experimental evidence that the elaboration of distal dendrites of abGCs depends on sensory input (*Saghatelyan et al., 2005*; *Yoshihara et al., 2012*; *Yoshihara et al., 2015*), we biased the dendritic field of abGCs such that abGCs were more likely to have potential synapses with MCs that were more active when the GC was 8–14 days old (*Figure 3A*). While this did not substantially affect flexibility or stability of the network (*Figure 3—figure supplement 1A*), it allowed the model to achieve strong memories even when the GCs could only connect to a small fraction of all MCs, consistent with the sparse connectivity of the bulb (*Figure 3B*). More importantly, it led to the preferential recruitment of abGCs that arrived in the OB and were still developing during enrichment (*Figure 3C*, blue areas) over those that were already in their critical period (*Figure 3C*, yellow area). This matches the experimental results, which show that these abGCs encode the memory (*Forest et al., 2020*; *Forest et al., 2019*).

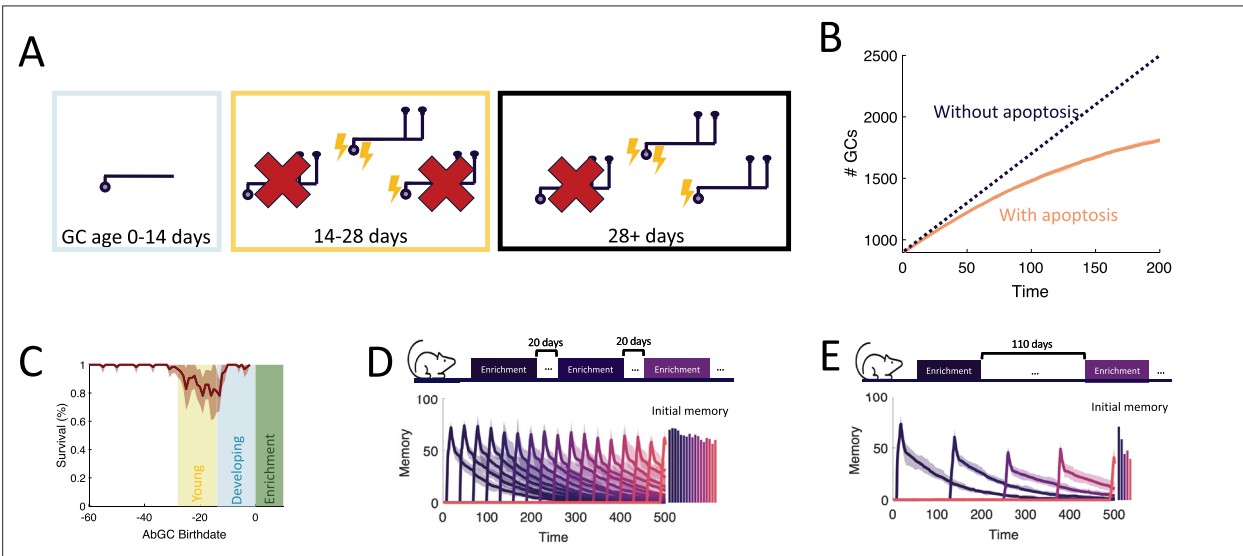

**Figure 4.** Apoptosis. (**A**) AbGCs in their critical period (14–28 days) require a higher level of activity to survive. (**B**) The growth of the GC layer over time. Line: mean across eight trials, shaded area: full range (**C**) The portion of surviving GCs as a function of birthdate. The shaded regions are as in *Figure 2E*. Line: mean across eight trials, shaded area: mean ± standard deviation. (**D, E**) Sequential enrichment simulations differing in the inter-enrichment interval. Each curve corresponds to the mean memory of a stimulus over eight trials, and the shaded areas show the range over all trials. The bar plots show the mean initial memory for each stimulus. Lines: mean across eight trials, shaded areas: full range.

The online version of this article includes the following figure supplement(s) for figure 4:

**Figure supplement 1.** Effects of increased abGC survival during enrichment.

**Figure supplement 2.** Retrograde interference.

We also examined the savings effect (*Ebbinghaus, 1885*) in which the re-learning of forgotten information occurs more rapidly than the initial learning. This effect has been observed in an olfactory associative learning task, where abGCs seem to be playing a significant role (*Sultan et al., 2010*). We examined this in our perceptual learning framework by simulating a similar relearning experiment and exploring what happens if neurogenesis is blocked during relearning (*Figure 3D*). The model predicts that GCs that initially encoded the memory are re-recruited to the odors during re-enrichment (*Figure 3E*). While these GCs were no longer as highly excitable or plastic as young GCs, they had the advantage that their dendritic fields overlapped strongly with MCs that are excited during enrichment, allowing them to efficiently re-establish their synaptic connections. This led to savings where the memory increased more rapidly during the re-enrichment (*Figure 3F*).

Additionally, despite the fact that neurogenesis is typically required for learning, the model predicts that savings will be seen even if neurogenesis is blocked during the second enrichment (*Figure 3E and F*). Importantly, these observations did not occur without the activity-dependent dendritic elaboration of young abGCs (*Figure 3—figure supplement 1B*), meaning the dendrites encoded 'latent memories' that facilitated the rapid re-expression of a previously encoded memory upon re-exposure to the stimulus.

## Targeted apoptosis maintains flexibility

The final property of GC development that we investigated is the death of GCs through apoptosis, which is heightened for abGCs in their critical period (*Figure 4A*). This critical period of survival led to two noteworthy phenomena. First, when tracking the growth of the OB, the number of GCs initially grew approximately linearly, while growth eventually started to slow down (*Figure 4B*), as observed experimentally (*Platel et al., 2019*). Second, there was a high rate of apoptosis in older young abGCs (*Figure 4C*, yellow area) in response to olfactory enrichment but not in abGCs that arrived in the OB during enrichment (blue area) or mature GCs (unshaded area). This is similar to experimental results that show enhanced death of 'middle-aged' abGCs, but not young or more mature abGCs (*Mouret et al., 2008*).

What are the potential implications of apoptosis on memory encoding? One known outcome is that it subjects newly formed memories to retrograde interference (*Forest et al., 2019*). In these experiments, there were two enrichment periods with odors that activated largely non-overlapping sets of GCs. If the second enrichment occurred while abGCs that encoded the memory of the first enrichment were still in their critical period, then these abGCs succumbed to apoptosis and the memory was extinguished. Our model captured this and identified that the dissimilarity of the odors is essential (*Figure 4—figure supplement 2*): because the connectivity developed by the odor-encoding GCs during the first enrichment was odor-specific, these GCs were inactive during the second enrichment and therefore more susceptible to apoptosis. If, alternatively, there was a long enough gap between the enrichment periods, or if the odors from the first enrichment were present during the second enrichment, then the abGCs encoding the first memory survived and the memory was consolidated in both the model and experiments. This highlights how apoptosis can selectively remove abGCs, and how only naive, young abGCs are available to encode new odors.

Yet the question still remains whether apoptosis can facilitate the formation of new memories rather than exclusively eliminate old ones. Based on the observation in *Figure 2G* that the growing inhibition from persistent neurogenesis restricts the ability to form new memories, we hypothesized that apoptosis may help maintain the flexibility of memory formation. To test how well the network learned under different levels of apoptosis, in *Figure 4D and E* the network was enriched with a variable frequency of enrichments, with each new enrichment consisting of a new, sparse, random pair of odors (*Figure 4—figure supplement 1A*). Because enrichment eliminated a large portion of abGCs that were late in their critical period at enrichment onset (*Figure 4C*), we expected to see more apoptosis and thus improved learning in trials with greater enrichment frequency. Indeed, with high frequency, the network flexibly learned all odors similarly well (*Figure 4D*), but with low frequency, the initial memory strength of later odors decreased substantially as neurons accumulated in the OB (*Figure 4E*, *Figure 4—figure supplement 1*). As a result, the model predicts that long periods without olfactory enrichment or with apoptosis blocked negatively impact the ability to learn new odors.

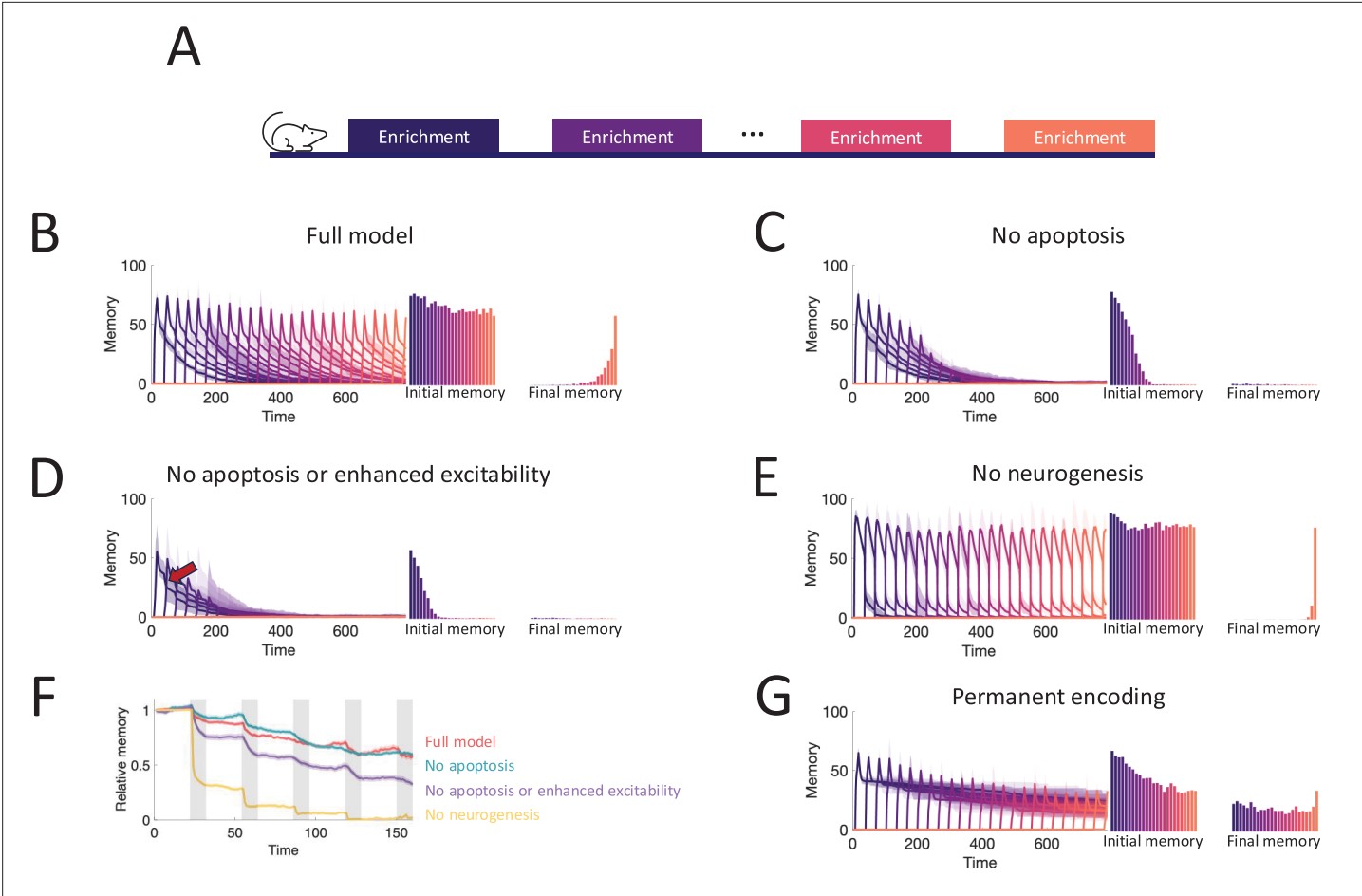

**Figure 5.** Neurogenesis for life-long learning. (**A**) Protocol for sequential enrichment simulations. (**B–E**) Memory evolution of full model, model without apoptosis, model without apoptosis or age-dependent excitability, and model without neurogenesis, respectively. Each curve corresponds to the memory of a different set of stimuli. First bar plot plots show the mean initial memory, while the second bar plot shows the mean memory at the end of the simulation. Lines: mean across eight trials, shaded areas: full range. In (**E**), $p = 0.15$ and $R_0 = 0.003$ so that the model learns and forgets at a rate similar to that of the full model. (**F**) Memory of the first enrichment in (**B-E**) relative to the memory of that same enrichment without subsequent enrichments. Lines: average across trials, shading: standard error of the mean. Shaded areas: times when the model is exposed to a new stimulus. (**G**) Memory for sequential enrichment if the connectivity of mature abGCs is frozen. Lines: mean across eight trials, shaded areas: full range.

The online version of this article includes the following figure supplement(s) for figure 5:

**Figure supplement 1.** GC population size over time.

## Adult neurogenesis supports life-long learning

To study the capacity of the OB to continually encode new stimuli, we simulated an experiment where we sequentially enriched the OB network on 25 sparse, random odor pairs (*Figure 5A*) and measured how the properties of the memories change over time. Our full model of the OB supported the flexible encoding of stable memories (*Figure 5B*). Learning flexibility, measured as the initial memory of the enrichment odors, was strongest at the start of the simulation; but it declined only slightly in subsequent enrichments and approached a steady value when the growth of the network started to saturate (*Figure 5—figure supplement 1*). Additionally, memories remained stable: the decay of the individual memory traces was barely affected by subsequent, interfering enrichments. This resulted in multiple odors having substantial memories at the final time of measurement, and these memories were graded according to the time of acquisition.

To show what properties of the OB are required to produce these results, we repeated the simulations under several different model conditions. First, in *Figure 5C*, we blocked apoptosis. While memories still remained stable, flexibility suffered greatly as the network eventually failed to learn. Notably, the memories of early enrichments were not strongly impacted, indicating that short-term

suppression of apoptosis should not affect learning, consistent with observations (*Mouret et al., 2009*). Next, we additionally removed the enhanced excitability of young abGCs (*Figure 5D*). Not only did flexibility decline more substantially, but memory stability also suffered as larger drops (see arrow) were evident in the individual memory traces during later enrichments. Finally, we considered a non-neurogenic network featuring only synaptic plasticity and none of the transient properties of abGCs (*Figure 5E*). While this network could flexibly form new memories, they were not stable as each subsequent enrichment led to the overwriting of old memories, severely limiting the memory capacity.

To evaluate quantitatively how the memory that was formed in the first enrichment deteriorated due to the ongoing encoding of new odors, we compared the memory traces of the first enrichment in *Figure 5B–E* to the memory trace of the same odor, but without any subsequent enrichments. Taking the ratio between these two traces shows the extent to which new stimuli overwrote old memories (*Figure 5F*). The non-neurogenic model displayed the most overwriting, with prominent overwriting events during the first few enrichments. Next, the network without the transiently enhanced excitability of young abGCs featured a moderate amount of overwriting, as young abGCs were no longer preferentially recruited to learn new odors. Finally, the full model and the model without apoptosis featured similarly low levels of overwriting, indicating that apoptosis is required for memory flexibility but not stability.

Because of the low degree of overwriting occurring in our full model of the OB, memory decay was primarily due to spontaneous synaptic changes. A large degree of spine turnover has been observed at MC-GC synapses in response to spontaneous activity alone, even on mature GCs (*Livneh and Mizrahi, 2012*; *Sailor et al., 2016*). We investigated the implications of this by comparing the results of the full model (*Figure 5B*) to those of a model of the OB in which plasticity was instead completely prevented in older GCs (*Figure 5G*). In this model, new memories briefly decayed due to spontaneous plasticity in young abGCs but then were maintained for the duration of the simulation. Although there was explicitly no synaptic overwriting, the memories still featured a slow decay due to the apoptosis of odor-encoding GCs. Apoptosis was more prominent in this network because the odor-encoding neurons featured a larger number of synapses (*Figure 5H*) that would otherwise be reduced through spontaneous plasticity. Thus, there was a greater degree of inhibition in the network, reducing GC survival (*Figure 5—figure supplement 1*). Importantly, these excess synapses led to more interference, which degraded the flexibility in this network (initial memories in *Figure 5B and G*). This suggested that the strong synaptic fluctuations among mature GCs served to increase flexibility by removing potentially interfering, unmaintained memories of odors. Of course, this strategy has the drawback that memories are less stable, but this drawback is mitigated by the latent memories of the odors being stored in the dendrites of abGCs, which allow for their rapid re-acquisition if the odors are once again present in the environment (*Figure 3*).

## Discussion

Here, we showed a clear computational advantage of adult neurogenesis in ongoing memory encoding. In a computational model of the OB, the increased plasticity exhibited by abGCs in their critical period allowed them to rapidly encode new information, while their subsequent development and the resulting decrease in plasticity rate ensured that the memories they encode remain stable. Meanwhile, the increased excitability of young abGCs enhanced their preferential recruitment in learning new information, while also helping maintain the flexibility of the system as new neurons accumulate in the OB. Apoptosis similarly helped maintain learning flexibility through the targeted removal of abGCs that fail to learn the odors presented during their critical period. Furthermore, the activity-dependent dendritic elaboration of juvenile abGCs led to pre-configured sub-networks of similarly aged abGCs that enabled the rapid re-acquisition of memory. All these elements were required to reproduce the relevant experimental results.

### Other models addressing the flexibility-stability problem

From a signal-theoretic perspective, both initial strength and duration of a new memory should improve as synapses are added to the network. Due to the metabolic cost of forming and removing synapses, it is important to use them efficiently. This is especially true for neurogenic systems in which

the increase in the number of synapses is associated with the addition of new neurons. The efficiency of the system can be characterized by how strongly its memory performance increases when the number $N$ of synapses is increased.

To assess this efficiency, we adapted a framework proposed by *Fusi et al., 2005*; *Roxin and Fusi, 2013* to analyze how memories degrade in response to ongoing plasticity (Appendix 2). In systems where both learning and forgetting occur on the same, fast time scale, the overall memory capacity grows only logarithmically with $N$ (*Amit and Fusi, 1994*; *Fusi and Abbott, 2007*). However, our computational model predicted that a network with age-dependent plasticity rates can have a vastly larger memory capacity on the order of $\sqrt{N}$ (Appendix 2). This emphasizes not only that the augmented capacity of the model predominantly arises from the transiently increased plasticity rather than the mere addition of synapses, but also that this model efficiently uses the new synapses provided by adult neurogenesis.

These results are commensurate with earlier models specifically designed to address the flexibility-stability dilemma, particularly the classical cascade model (*Fusi et al., 2005*) and the partitioned-memory model of systems consolidation (*Roxin and Fusi, 2013*; *Appendix 2—figure 1*). However, our model does not yield an increase in memory duration that is linear in $N$ as demonstrated by the more complex bidirectional cascade model (*Benna and Fusi, 2016*). This difference underscores the intricate interplay of synaptic plasticity mechanisms and their impact on memory consolidation.

## Why neurogenesis?

If there exist alternative methods that are theoretically demonstrated to consolidate memory with comparable effectiveness (*Fusi et al., 2005*; *Roxin and Fusi, 2013*; *Benna and Fusi, 2016*), why does the OB opt for adult neurogenesis? Adult neurogenesis is a rare phenomenon in mammals, but more common in organisms with less complex nervous systems, such as reptiles, birds, and fish. This contrast suggests an evolutionary pressure to reduce neurogenesis that is resisted by the OB (*Aimone, 2016*). To understand why neurogenesis is nevertheless favored in the OB, it is crucial to examine the specific function of the OB and its constituent GCs.

There have been numerous studies that propose different theories for the function of the MC-GC network in processing odors. In particular, both processes implementing Bayesian inference (*Grabska-Barwińska et al., 2017*; *Tootoonian et al., 2022*) and sparse incomplete representations (*Koulakov and Rinberg, 2011*) have been proposed, with each offering a computational rationale for how the olfactory bulb (OB) might achieve pattern separation — an effect that has been experimentally observed in the OB (*Gschwend et al., 2015*). This pattern separation is supported by recurrent inhibition between MCs that is mediated by GCs (*Wick et al., 2010*; *Wiechert et al., 2010*; *Koulakov and Rinberg, 2011*). This processing is similar to contrast enhancement and edge enhancement that is performed in visual processing already in the retina. Although the OB and retina differ in their anatomical organization (e.g. the retina lacks the reciprocal connectivity found in the OB), both systems employ lateral inhibition between neurons with correlated activity. In natural visual stimuli, correlations are high among neighboring 'pixels', reflecting the contiguity of physical objects. For such stimuli, local lateral inhibition among neighboring 'pixels' is sufficient (*Hartilne and Ratlife, 1957*) and the connectivity does not need to adapt to specific visual scenes.

Olfactory stimuli, however, lack this topographic correlation structure and are extremely high-dimensional. Thus, even if similar odors were to differ only in neighboring 'olfactory pixels' in this high-dimensional space, their projection onto the two-dimensional arrangement of glomeruli results in 'fragmented' activation patterns, wherein the relevant 'olfactory pixels' are widely distributed across large portions of the bulbar surface (*Cleland and Sethupathy, 2006*; *Soucy et al., 2009*). While initial contrast enhancement can then still be performed locally (*Cleland and Sethupathy, 2006*), computations that aim to incorporate the correlation structure of the stimuli require specific lateral connectivity over larger distances, as provided by GCs.

Only a subset of odors is innately relevant to a species and can likely be processed with a fixed, pre-wired connectivity. However, many odors are not experienced for the first time until later in life and must then be learned on demand. For instance, mother sheep encounter the odors that allow them to recognize their offspring only after that offspring is born (*Brennan and Keverne, 1997*). The OB must therefore allow for lifelong learning, where the flexible learning of new stimuli while maintaining the stability of old memories is crucial. Our modeling demonstrates that neurogenesis and the

development of abGCs provide exactly this. However, we also showed that both the survival of too many memories and of too many GCs can harm the flexibility of the network. This raises the question of which memories and GCs should endure and for how long. Previous experiments and our modeling show that after an odor is learned, the abGCs recruited to that odor are subject to retrograde interference during their critical period: unless the initial odor is maintained in the environment, the learning of a new memory decreases the survival of the abGCs associated with the earlier odor (*Forest et al., 2019*).

Past their critical period, the abGCs are much more stable. Nevertheless, the animals lose the memory of the learned odor over the course of 30–40 days (*Forest et al., 2019*). However, at that point, they can re-acquire that memory faster than they had learned it initially (*Sultan et al., 2010*). This is consistent with the relevant synaptic connections being lost, reflecting the strong spontaneous formation and removal of GC spines (*Sailor et al., 2016*; *Meng and Riecke, 2022*), while the relevant GCs are still present and provide - through their stable dendrites (*Mizrahi, 2007*; *Sailor et al., 2016*) - a latent memory that can quickly be reactivated by reforming the relevant synapses.

While experiments have shown that cortical feedback and neuromodulation in response to different environmental or behavioral states may provide an apoptotic signal (*Yokoyama et al., 2011*; *Komano-Inoue et al., 2014*), what precisely controls the survival of GCs is not quite clear. In *Sultan et al., 2010*, the GCs die over the course of 90 days in the absence of the odor they memorized. In the experiments of *Platel et al., 2019*, however, in which animals were not exposed to any tasks, little if any cell death is reported. From a functional point of view, the long-term survival of odor-encoding abGCs in the absence of further exposure to the memorized odor is expected to be controlled by the need to avoid interference with new odors while accommodating the possibility that the same odor will reappear at some later point in time.

Thus, our modeling suggests that the high dimension of odor space, together with the need for animals to learn specific novel odors quickly but stably, strongly favors structural plasticity through adult neurogenesis and apoptosis.

## Relation to adult neurogenesis in the hippocampus

Adult neurogenesis also occurs in granule cells in the hippocampus (*Christian et al., 2014*; *Kempermann et al., 2015*). Like olfactory abGCs, hippocampal abGCs exhibit transiently increased plasticity and excitability (*Lledo et al., 2006*); however, they are excitatory and constitute the principal neurons of their network. Hippocampal GCs contribute to pattern separation and memory acquisition much like olfactory GCs, but also play an important role in other aspects of spatial and contextual memory (*Aimone et al., 2009*; *Christian et al., 2014*).

On the topic of memory stability, recent experiments have shown that up-regulating hippocampal neurogenesis can enhance the forgetting of previously learned information over the course of a month, while down-regulating it can diminish the forgetting (*Akers et al., 2014*). This suggests that interference from new cells makes old memories unstable and aids in memory clearance (*Akers et al., 2014*; *Epp et al., 2016*; *Gao et al., 2018*; *Guskjolen and Cembrowski, 2023*). Computational modeling and experiments have suggested that this forgetting may specifically be happening at the mossy fiber-CA3 synapse (*Tran et al., 2019*; *Guskjolen and Cembrowski, 2023*).

In our model, while abGCs born after learning cause interference in the network and perturb MC responses, only a vast increase of the post-learning neurogenesis rate would significantly alter the memory duration (*Appendix 3—figure 1*). Our research suggests a different role for adult neurogenesis: the age-dependent properties of the abGCs can stabilize memories for more than a month that would otherwise decay over the course of a week (*Figure 2B*). In view of the metabolic costs of neurogenesis, this would seem to be a better investment than memory clearance.

## The development of birthdate-dependent subnetworks

A key outcome of our model is the development of birthdate-dependent odor-specific subnetworks. This applies not only to the synaptic networks, but also to the underlying dendritic networks. In the model, this is because abGCs born in a similar time window begin development in a similar environment. We therefore predict that the rapid re-learning observed in an olfactory associative learning task (*Sultan et al., 2011*) would also occur in a perceptual learning experiment and would still be present even if neurogenesis was blocked after the initial enrichment.

This would be notable for two reasons. First, it has been shown that neurogenesis is required for perceptual learning of fine odor discrimination (*Moreno et al., 2009*). Therefore, if re-learning were to occur without neurogenesis, then it would indicate that there is some structure storing a latent memory that is not expressed behaviorally. Second, the fast re-learning was not seen in the model without activity-dependent dendritic elaboration, so it would suggest that the dendritic tree may be a substrate of this latent memory. Importantly, these latent memories only persist as long as the neurons encoding them survive. It remains to be seen if periodic re-exposure to stimuli after learning can extend the lifetime of odor-encoding abGCs, as would be expected in a model where the OB predominantly eliminates GCs that encode extraneous information.

Similar results of birthdate-dependent subnetworks have been observed experimentally as a result of embryonic neurogenesis in the hippocampus (*Huszár et al., 2022*). In this study, place cells in CA1 were observed to form assemblies where neurons were more likely to be in the same assembly with other neurons born on the same day compared to those born on different days. Importantly, in a place alternation task, these cells have also been observed to remap together, maintaining sub-assemblies across environments. In this sense, the hippocampal neurons exhibited a set of pre-configured activity patterns dependent on their birth date, reminiscent of the latent memories we describe in our model.

## The role of apoptosis in learning

In addition to predictions about relearning, the model predicts that apoptosis helps maintain the flexibility of the OB and that reduced apoptosis would lead to memory deficits (*Figures 4 and 5C*). In standard, non-enriched laboratory conditions, the observed rate of apoptosis of abGCs after they have established themselves in the OB network is low (*Platel et al., 2019*). In such conditions, the model predicts that abGCs that fail to encode any relevant information accumulate in the OB and add non-specific inhibition to the OB, making it more difficult for new abGCs to integrate into the OB when new odors are presented.

Olfactory enrichment eliminates many abGCs that are late in their critical period that may otherwise survive (*Forest et al., 2019*; *Mouret et al., 2008*). At the same time, it also enhances the number of abGCs that survive until they start integrating into the network, despite the unchanged proliferation rate (*Rochefort et al., 2002*). The latter mechanism has not been built into the model, making it natural to wonder if this would impact the results in *Figure 4*. To address this, we doubled the number of new neurons available to integrate into the network during enrichment (*Figure 4—figure supplement 1*) and found this did not qualitatively change the results.

We expect that the enhanced flexibility due to apoptosis would likely be most pronounced in mice between six and twelve months, when olfactory perceptual memory deficits start to appear (*Greco-Vuilloud et al., 2022*) and the growth of the granule cell layer starts to slow down (*Platel et al., 2019*). Indeed, very recently, it has been observed that long-term olfactory enrichment improves memory in this cohort of mice (*Terrier and Greco-Vuilloud, 2024*).

Importantly, there are many other modulators of abGC survival beyond olfactory enrichment. For example, apoptosis can be induced by a variety of behavioral states (*Yokoyama et al., 2011*; *Yamaguchi et al., 2013*; *Komano-Inoue et al., 2014*). In the natural world, we would therefore expect to see a more substantial degree of apoptosis, which the model predicts to preserve the flexibility of the OB.

## Model assumptions and outlook

In developing the computational model, we have made a number of assumptions that are consistent with current experimental observations, but for which the underlying biophysical mechanisms are still poorly understood. The model, therefore, points to aspects of neurogenesis, the experimental exploration of which would be particularly important in order to understand the functional relevance of adult neurogenesis.

We have assumed that MC-GC connectivity is fully reciprocal; however, this is not the case in the biological system (*Egger and Kuner, 2021*). Previous work from our group suggests that the result presented here would be robust to moderate drops in reciprocity (*Chow et al., 2012*). Additionally, an important aspect of olfactory processing is the spike timing of MCs (*Wilson et al., 2017*; *Bolding and Franks, 2017*; *Chong et al., 2020*). The relative timing of spikes from a few select MCs can have an outsized effect on olfactory perception (*Chong et al., 2020*). Capturing these timing

aspects would require more complex spiking models (*Gilra and Bhalla, 2015*; *Li and Cleland, 2017*), rather than simple rate models. Notably, it has been shown that MC-GC connections alter MC spike timing and phase information (*Gilra and Bhalla, 2015*; *Egger and Kuner, 2021*), highlighting connectivity as a key quantity in the quantification of memory. This motivates our ideal observer approach, which focuses on network connectivity rather than neuronal activity. Depending on how the memory is represented in the neuronal activity, only a subset of the connectivity may be needed to support (*Gilra and Bhalla, 2015*) a given memory. Assuming that learning novel unrelated odors will overwrite the learned connectivity randomly, that subset of the connectivity is expected to persist or deteriorate in proportion to the memory strength we associate with the full connectivity. We therefore expect that our modeling captures the essentials of the memory persistence, independent of how odors are encoded in neuronal activity.

A prominent feature of olfaction is the extensive top-down input to the OB, which can be through glutamatergic input to specific bulbar neurons (*Boyd et al., 2015*; *Otazu et al., 2015*; *Wu et al., 2020*), or can act diffusely via neuromodulators (*Mandairon et al., 2006*). These inputs can provide additional information to the OB about the context of the sensory input (*Mandairon et al., 2014*), its valence (*Lindeman et al., 2024*), or the internal state of the animal (*Boyd et al., 2015*). They can arise from the piriform cortex or the anterior olfactory nucleus and modulate mostly the activity of mitral or tufted cells (*Chae et al., 2022*).

In the context of memory formation, a key aspect is whether this additional input can prioritize certain stimuli over others. This could occur through a top-down reward signal for a specific odor or through neuromodulatory input signifying something akin to attention or novelty (*Veyrac et al., 2007*). In this case, the top-down input could modify the learning process in an odor-specific way to protect the memory of important stimuli against overwriting by memories of less important stimuli.

In this study, however, we are concerned with perceptual learning where animals learn to distinguish between similar stimuli through repeated exposure, but without any cues indicating the relative importance of each stimulus. Notably, although no explicit reinforcement is provided, this form of learning has been shown to require noradrenaline (*Veyrac et al., 2007*; *Veyrac et al., 2009*), in contrast to associative learning which can occur independently of noradrenergic signaling (*Veyrac et al., 2009*). Based on this, in our model, we assume that the noradrenergic system is active during the learning process, but that it does not differentiate between stimuli. This could be implemented as a stimulus-independent bias in neural and synaptic properties, and our current model should suffice regardless of the specific implementation. In this case, the top-down input would be the same for all stimuli in question, and the system would still face the stability-flexibility dilemma.

If, however, specific stimuli are associated with specific non-olfactory contexts, for example visual inputs (*Mandairon et al., 2014*), top-down inputs could mark some stimuli as more important to retain. Such top-down input would need to be modeled explicitly. An earlier computational model of adult neurogenesis suggests that in such situations, the GCs develop odor-specific receptive fields with respect to their sensory and their top-down inputs (*Adams et al., 2019*). As a result, the top-down inputs associated with a given stimulus selectively activate GCs that, through their reciprocal connections, inhibit predominantly MCs coding for that stimulus. This gives the top-down inputs very specific control of the bulbar response to the associated odor and enhances or reduces odor discrimination depending on the non-olfactory input.

The combined neurogenic and synaptic plasticity investigated in the current paper leads to a bulbar network structure that is similar to the bulbar component generated by the top-down model (*Adams et al., 2019*). A straightforward extension of the combined model along the lines pursued in *Adams et al., 2019* would most likely also reveal specific cortical control of bulbar odor processing by top-down inputs, which could include modification of the learning process depending on non-olfactory information transmitted by the top-down input, as observed in *Wu et al., 2020*. Such feedback might not only alter the plasticity of existing circuits but also bias the recruitment of abGCs toward particular stimulus features, thereby shaping the composition of the resulting engram. Indeed, different learning paradigms have been shown to recruit distinct cohorts of abGCs (*Mandairon et al., 2018*; *Wu et al., 2020*), suggesting that task structure directly influences how and where information is stored. However, when the different stimuli in question do not differ in their contextual or valence information, as is the case in basic perceptual learning, the system still has to deal with the stability-flexibility dilemma. While it is expected that the age-dependent properties of

**Table 1.** Age-dependent parameters.
GCs were considered immature if they were added to the network within 14 time steps, corresponding to their critical period.

| | Critical period | Fully mature | Description | Usage |
|---|---|---|---|---|
| $p$ | 0.2 | 0.02 | Plasticity rate | Scales α, β, $R^+$, $R^-$ |
| $r$ | 1.5 | 1 | Excitability of GCs | *Equations 2, 6* |
| $G_0$ | 0.5 (enriched), 0 (unenriched) | −0.45 | Survival threshold | *Equation 11* |

the abGCs will alleviate that issue also in this more complex scenario, answering this question definitively would require further studies.

## Materials and methods
### Neuron model
We model the activity of MCs and GCs within a firing-rate framework,

$$\tau_M \frac{dM_i}{dt} = -M_i + [S_i - \gamma \sum_j w_{ij} G_j]_+,$$

(1)

$$\tau_G \frac{dG_i}{dt} = -G_i + r \sum_j w_{ji} M_j.$$

(2)

Here, $M_i$ and $G_i$ represent the firing rates of individual mitral cells and granule cells, respectively, and $[x]_+$ denotes a threshold-linear rectifier: $[x]_+ = x$ for $x > 0$ and $[x]_+ = 0$ for $x \leq 0$. The excitatory input to MC $i$ consists of the term $S_i$ (see Stimulus model).

The synaptic weights $w_{ij}$ are 1 if the synapse between MC $i$ and GC $j$ is fully functional and 0 otherwise. Note that we assume each synapse is reciprocal, $w_{ij} = w_{ji}$. The strength of the inhibition by granule cells is then given by $\gamma$. The parameter $r$ captures the excitability of granule cells. Throughout this study, we set the number of MCs $N_{MC}$ as 225, and the initial number of GCs $N_{GC}$ as 900.

To simulate neurogenesis, at each time step $N_{add}$ GCs were added to the synaptic network. We chose $N_{add}$ to be 8 so that the ratio of new cells to existing cells would be consistent with experimental estimates (*Kaplan et al., 1985*). These new neurons had dendritic spines and were immediately capable of providing inhibition, corresponding in mice to abGCs that are about 14 days old (*Carleton et al., 2003*; *Kelsch et al., 2008*). To reflect the observation that young abGCs, aged 14–28 days, are more excitable than mature GCs, we made the parameter $r$ age-dependent. For simplicity, we assumed this age dependence followed a step function, such that young cells had a high level of excitability and mature cells had lower ones (*Table 1*).

### Network structure
Because a GC can only form synapses with MCs whose dendrites come physically close to its own dendrites, we impose the restriction that GCs can only make synapses with a predetermined set of $N_{conn}$ MCs. Since the MC dendrites extend across large portions of the olfactory bulb, we allowed connections between cells independent of the physical distance between somata. For neonatal GCs, this subset was randomly chosen. For abGCs, however, starting in Section 'The dendritic structure of abGCs latently encodes memories', this subset was chosen in a semi activity-dependent manner to reflect the activity-dependent and -independent mechanisms which guide dendritic growth in developing cells (*Wong and Ghosh, 2002*; *Saghatelyan et al., 2005*; *Dahlen et al., 2011*; *Yoshihara et al., 2012*). To this end, we calculate a variable, $\tilde{M}_{i,j}$ that is used to determine which MCs $i$ a given abGC $j$ can connect to:

$$\tilde{M}_{i,j} = [\bar{M}_i - \theta_M]_+ + \varepsilon_{i,j}.$$

(3)

Here, $\bar{M}_i$ is the average activity of MC $i$ over the 6 days preceding the addition of GC $j$ to the network (corresponding to the amount of time between when an abGC arrives in the OB and when its starts spiking *Carleton et al., 2003*), $\theta_M$ is the threshold of activity required to induce dendritic

growth, and $\varepsilon_{i,j}$ is a random variable that mimics the complex structure of the MC dendrites and a random position of the GC soma relative to the set of MCs when it starts developing its dendritic arbor.

Whenever a GC $j$ is added to the network, the set $\{MC\}_j^{(pot)}$ of MCs to which it can connect is given by the $N_{conn}$ MCs that have the highest $\tilde{M}_{i,j}$ values at that time. Once a GC is added to the model, its set $\{MC\}_j^{(pot)}$ does not change; this is to reflect that dendrites of abGCs are relatively stable once the abGCs start spiking (*Mizrahi, 2007*; *Sailor et al., 2016*). Lastly, when GC $j$ is added to the network, it makes functional synapses with $N_{init}$ MCs, randomly chosen from $\{MC\}_j^{(pot)}$. This applies to both neonatal and adult-born GCs.

## Synaptic plasticity

We model synaptic dynamics as a Markov chain with three states: non-existent, unconsolidated, and consolidated. We include the unconsolidated state, since experimentally it is found that a substantial fraction of spines that are identified optically is lacking PSD-95 (*Saha et al., 2021*). We assume state transitions from non-existent to unconsolidated occur randomly with a constant rate α and state transitions from unconsolidated to non-existent with constant rate $\beta$. Meanwhile, state transitions to and from the consolidated state rely on pre- and post-synaptic activity. This assumption is supported by experiments as it has been shown that GC spine dynamics depend on GC activity (*Breton-Provencher et al., 2016*; *Breton-Provencher et al., 2014*; *Saha et al., 2021*). Moreover, the formation and removal of consolidated synapses appear to be separate processes that depend on the calcium concentration at the synapse (*Kasai et al., 2021*; *Stein et al., 2021*; *Park et al., 2022*). Therefore, we express the rate $R_{i,j}^+$ for the consolidation of an unconsolidated synapse between MC $i$ and GC $j$ and the corresponding deconsolidation rate $R_{i,j}^-$ as functions of a variable $[Ca]_{i,j}$ that mimics the calcium concentration at the synapse,

$$R_{i,j}^+ = R_0 + \frac{1}{2}(\tanh(g([Ca]_{i,j} - \theta_j^+)) + 1) \tag{4}$$

$$R_{i,j}^- = R_0 + \frac{d}{\cosh(g([Ca]_{i,j} - \theta_j^-))}. \tag{5}$$

$R_0$ is the rate of spontaneous spine changes, $g$ is a constant, and $\theta_j^+$ and $\theta_j^-$ are parameters related to the thresholds of spine formation and removal specific to GC $j$. Additionally, $d$ is the relative rate of deconsolidation to consolidation, which we set to be less than one to reflect that consolidation is faster than deconsolidation (*Kasai et al., 2021*).

The functional forms of these equations were chosen to qualitatively resemble those of the Artola-Bröcher-Singer (ABS) rule of synaptic plasticity (*Artola et al., 1990*; *Artola and Singer, 1993*) where highly active unconsolidated synapses are more likely to undergo consolidation (*Vardalaki et al., 2022*) and less active consolidated synapses more likely to undergo deconsolidation (*Kasai et al., 2021*). We further assume that each synaptic state is associated with a fixed weight value. Specifically, non-existent and unconsolidated synapses have synaptic weight zero, while consolidated synapses have weight one. We recognize that this approach ignores synaptic weight plasticity, but note that these binary synapses are representative of a class of realistic synaptic models (*Fusi, 2021*). Moreover, only limited information is available for the weight plasticity of MC-GC synapses (*Gao and Strowbridge, 2009*).

To model the local calcium concentration $[Ca]_{ij}$ at the synapse between MC $i$ and GC $j$, we adapt the model presented by *Graupner and Brunel, 2012* to our firing rate framework:

$$\tau \frac{d[Ca]_{ij}}{dt} = -[Ca]_{ij} + C_{pre}rM_i + C_{post}G_j. \tag{6}$$

Here, $C_{pre}$ captures calcium influx driven by pre-synaptic activity: glutamatergic input from MC $i$ and the resulting depolarization within the spine of GC $j$ allow calcium influx through NMDARs and voltage-gated calcium channels. $C_{post}$ captures calcium influx driven by post-synaptic activity that is independent of glutamate release from MC $i$: depolarization in the spine that is driven by (global) spikes in the GC dendrite allows calcium influx through voltage-gated calcium channels. The global spikes are reflected in the activity of GC $j$.

Next, we transform the rates $R_{i,j}^+$ and $R_{i,j}^-$ into state-transition probabilities through the functions

$$P_{ij}^+ = 1 - \exp(-R_{i,j}^+), \tag{7}$$

$$P_{ij}^- = 1 - \exp(-R_{i,j}^-), \tag{8}$$

and stochastically consolidate synapses with probability $P_{ij}^+$ and deconsolidate synapses with probability $P_{ij}^-$.

In order to maintain stability of the network, we impose a sliding threshold rule on the local consolidation parameters $\theta_j^\pm$,

$$\theta_j^+ = \max(\theta^+, k[\bar{Ca}]_j), \tag{9}$$

$$\theta_j^- = \max(\theta^-, k[\bar{Ca}]_j), \tag{10}$$

where $\theta^+$ and $\theta^-$ are the minimal thresholds for consolidation and deconsolidation, respectively, $k$ is a parameter, and $[\bar{Ca}]_j$ is the calcium concentration averaged across GC $j$. The sliding threshold represents intracellular competition between synapses.

Lastly, we scale all synaptic transition rates $\alpha$, $\beta$, $R^+$ and $R^-$ by a common plasticity rate $p$ of the GC. To reflect the age-dependence of the plasticity rate, we make this parameter dependent on the age of each individual GC (see *Table 1*). The values of this parameter were chosen to match experimental data in *Sailor et al., 2016* that measure spine turnover in GCs of different ages.

## Apoptosis

We model apoptosis as an activity-dependent process where neurons are removed stochastically with probability given by the sigmoid

$$\mathcal{P}_i = \frac{1}{2}(1 - \tanh{(5(G_i - G_0))}). \tag{11}$$

Here, $\mathcal{P}_i$ is the apoptotic probability of granule cell $i$ and $G_0$ is an age-dependent survival threshold (*Table 1*). It reflects the fact that GCs are more susceptible to apoptosis in their critical period of survival (*Yamaguchi and Mori, 2005*), but still allows a small chance of apoptosis in mature GCs as observed experimentally (*Yokoyama et al., 2011*). We model this by choosing the survival threshold $G_0$ to be higher for the immature abGCs than for the mature abGCs.

Recently, however, the degree of apoptosis has become controversial as it has been revealed that in standard conditions without any olfactory stimuli or behavioral task, there is in fact very little observed apoptosis (even for abGCs during their critical period; *Platel et al., 2019*). One potential explanation is that apoptosis is modulated by environmental and behavioral factors. Keeping with this, experiments have shown, for example, that survival of abGCs can be regulated by noradrenergic mechanisms in response to novel stimuli (*Veyrac et al., 2009*). It has also been shown that apoptosis is more commonly observed during certain behavioral states (*Yokoyama et al., 2011*; *Yamaguchi et al., 2013*; *Komano-Inoue et al., 2014*), and, when triggered, is enhanced in GCs receiving fewer sensory inputs (*Yokoyama et al., 2011*). Moreover, GC survival can be increased by increasing the intrinsic excitability of the cells and relies on NMDARs (*Lin et al., 2010*). Together, these results indicate that apoptosis depends on activity and age of GCs, as well as the environment and internal state of the animal. Therefore, to parsimoniously capture these results, we assume a 'removal signal' occurs during enrichment that can cause young abGCs to be even more susceptible to apoptosis, raising the $G_0$ value for young cells further (*Figure 1D*). This mechanism is similar to the two-stage model for GC elimination proposed by *Yokoyama et al., 2011*; *Yamaguchi et al., 2013*; *Komano-Inoue et al., 2014*.

## Stimulus model

The excitatory input to MC $i$ is given by sensory input $S_i^{odor}$ and a term $S^{spontaneous}$, through which the MCs have spontaneous activity even without sensory input,

$$S_i = S^{spontaneous} + S_i^{odor}. \tag{12}$$

We consider two different types of sensory input, $S_i^{odor}$. In the first, $S_i^{odor}$ is given by a mixture of two Gaussian activity profiles,

$$S_i^{odor} = \sum_k^2 A_k e^{\frac{(i - \mu_k)^2}{\sigma^2}} .$$

(13)

Here, $A_k$ is the amplitude of each stimulus, $\mu_k$ is the index of the greatest activated MC, and $\sigma^2$ is the width of the activity distribution. These stimuli can be visualized by sorting MCs according to their input (*Figure 2B*) and are interpreted as mixtures of two odorants. For the parameters we used, these stimuli broadly excited a large group of MCs, making them effective for visualizing the connectivity and GC recruitment in *Figures 2 and 3*.

For simulations involved in *Figures 4 and 5* where the model learns a large set of stimuli, we incorporate the known sparsity of odor-evoked glomerular activation patterns, using mixtures of sparse, non-overlapping binary stimuli,

$$S_i^{odor} = \sum_k^2 A_k I_k(i).$$

(14)

Here, $I_k$ is an indicator variable that is 1 if MC $i$ receives direct input for the stimulus and 0 otherwise. For each odor pair, or enrichment, we defined a new pair of indicator variables $I_k$ such that each one provides input to 10% of MCs (for example, see *Figure 4—figure supplement 1A*), a similar fraction to what has been observed experimentally (*Wachowiak and Cohen, 2001*). Although in the real system glomerular activation is not binary, we keep it as such for simplicity, having already shown with the Gaussian stimuli that the model can accommodate graded activation patterns. This results in denser stimulus representations and thus is a worst-case scenario for the model in terms of evaluating memory capacity (*Fusi, 2021*).

## Perceptual learning task

To evaluate the role of neurogenesis in olfactory learning, we simulated a protocol based on an implicit perceptual learning task that has been shown to require adult neurogenesis (*Moreno et al., 2009*). In this experiment, mice were presented with two similar odors for one hour a day over ten days, and they learned to discriminate between these odors implicitly through experience alone in the absence of reward or punishment. To this end, we simulate two epochs: a non-enrichment epoch where $S_i^{odor} = 0$ and an enrichment epoch where $S_i^{odor} \neq 0$ is picked at random from a set of enrichment odors. To reflect the sparser temporal properties of olfactory stimuli, we interleave stimulus presentations during the enrichment epoch where $S_i^{odor} = 0$. In addition, to reflect the slower timescales of neurogenesis and apoptosis, there were multiple stimulus presentations between time steps (referred to as 'days') where neurons were added and removed.

## Memory

To assess the ability of the model to learn, we introduce an anatomic memory measure that is based on the network connectivity. Using an ideal observer approach, we assume that we have access to all synaptic strengths in the network. While the brain is unlikely to use such specific information to express memories, this gives us a limit on memory strength and duration and allows us to analyze how the OB network changes in response to olfactory enrichment.

We characterize the memory strength $\mu_i^s$ with which odor pair $s$ is 'memorized' by GC $i$ by the similarity (scalar product) between the average activity of MCs in response to that odor pair and the inhibition levied on those MCs by a unit activation of the GC $i$,

$$\mu_i^s = \sum_{j=1}^{N_{MC}} w_{ji} \bar{M}_j.$$

(15)

Here, $\bar{M}_j$ is the mean activity of MC $j$ in response to both odors in the pair $s$. This reflects the fact that the plasticity processes of the model lead to a network connectivity that provides mutual inhibition between MCs reflecting their co-activity in response to the training stimuli. We use both odors in

the pair $s$ to characterize the memory, since, throughout this study, we examine how the network is able to learn to discriminate between two similar odors, which are both presented to the network in an alternating fashion.

We then define the total memory $\mu^s$ of odor pair $s$ in the network as

$$\mu^s = \sum_{i=1}^{N_{GC}} [\mu_i^s - \theta_\mu(i)]_+. \tag{16}$$

Here, $\theta_\mu(i)$ is a threshold describing the 'maximal null memory' of GC $i$. To obtain $\theta_\mu(i)$, we first determine the distribution of $\mu_i^s$ values for granule cell $i$ across a set of 10,000 reshuffled connectivities. Then we take $\theta_\mu(i)$ to be 3 standard deviations above the mean of this distribution. This provides us with a measure of how well the odor pair is encoded in the network above what would be expected from a random connectivity.

### Clustering analysis

To characterize learning-induced subnetworks within the OB, we performed hierarchical clustering using an agglomerative approach with Ward linkage on the columns of the connectivity matrix between MCs and GCs (*Pedregosa et al., 2011*). We then sought to identify the number of clusters present in the data using the resulting distances between groups of points returned by the algorithm. Due to the dependence of the clustering on the degree of the GCs, we did this using null distributions of the distances between groups found performing clustering on 10,000 shuffled networks, in the spirit of *Johnson et al., 2022*. This was done recursively. First, we compared the distance between

**Table 2.** Age-independent parameters.
Parameter values used in the simulation of the model unless stated otherwise.

| | Value | Description | Usage |
|---|---|---|---|
| $N_{MC}$ | 225 | Number of MCs | Network architecture |
| $N_{GC}$ | 900 | Initial number of GCs | Network architecture |
| $N_{add}$ | 8 | Number of GCs added each neurogenesis step | Network architecture |
| $N_{conn}$ | 30 | Number of MCs with which a GC can form synapses | Network architecture |
| $N_{init}$ | 10 | Number initial synapses made by each abGC | Network architecture |
| $\alpha$ | 1.5 | Rate of unconsolidated spine formation | Network architecture |
| $\beta$ | 1.5 | Rate of unconsolidated spine removal | Network architecture |
| $S^{spontaneous}$ | 0.1 | Spontaneous activity of sensory input | *Equation 1* |
| $\gamma$ | 0.004 | Inhibitory strength of GCs | *Equation 1* |
| $\theta^M$ | 0.15 | Threshold to induce dendritic growth | *Equation 3* |
| $\epsilon_{i,j}$ | $\sim U(0, 1.3)$ | Noise in dendritic formation | *Equation 3* |
| $R_0$ | 0.01 | Spontaneous rate of synaptic changes | *Equations 4, 5* |
| $g$ | 13 | Scaler | *Equations 4, 5* |
| $d$ | 0.7 | Relative rate of deconsolidation to consolidation | *Equation 5* |
| $C_{pre}$ | 3.6 | Presynaptic calcium contribution | *Equation 6* |
| $C_{post}$ | 1.1 | Postsynaptic calcium contribution | *Equation 6* |
| $\tau$ | 3.33 | Timescale of calcium dynamics | *Equation 6* |
| $k$ | 0.95 | Consolidation threshold scalar | *Equations 9, 10* |
| $\theta^+$ | 1.05 | Minimum threshold for consolidation | *Equation 9* |
| $\theta^-$ | 1 | Minimum threshold for deconsolidation | *Equation 10* |

the two largest clusters in the data with the null distribution of distances between the largest clusters of shuffled data. If the true distance was outside of the distribution of shuffled distances, then we deemed the cluster as significant and repeated the process on the two resulting subgroups. This continued until there were no new significant clusters.

## Robustness and parameters

The full list of parameters of the model and their default values is found in *Tables 1 and 2*. The parameters $C_{pre}$ and $C_{post}$ were fit using the genetic algorithm ga (*The MathWorks Inc, 2022*) to optimize memory following enrichment (*Appendix 3—figure 2A*). The parameters $\alpha$, $\beta$, and $p$ were fit to match the rates of spine turnover in young and mature abGCs found by *Sailor et al., 2016*. Meanwhile, other parameters were tuned by hand to be consistent with more coarse experimental evidence. For example, $R_0$ controls memory decay due to spontaneous synaptic changes and was chosen such that memories endure for up to 30 days (*Forest et al., 2019*). Since the forgetting may in part also be due to overwriting by other odors present, this value of $R_0$ may be somewhat too large. Additionally, the number of abGCs added each day, $N_{add}$ was chosen so that the ratio $\frac{N_{add}}{N_{CG}}$ matched experimental estimates (*Kaplan et al., 1985*). Still, several parameters had to be chosen without any available experimental support. Below is a discussion of a few selected parameters and their impact on the simulations.

The first parameters that we assessed were those associated with the abGC critical period (*Table 1*). The initial memory was robust to changes in the enhanced level of excitability of young abGCs, $r$, although learning declined slightly for the largest values we tested (*Appendix 3—figure 2B*). We next examined the model's dependence on the survival threshold of young abGCs, $G_0$ (*Appendix 3— figure 2G*). Increasing this parameter led to larger drops in the survival of abGCs that were already in their critical period at enrichment onset (yellow shaded area) without affecting the survival of most younger abGCs (blue shaded area), or the initial memory formed during enrichment.

We next explored the ramifications of other parameters associated with adult neurogenesis, starting with the neurogenesis rate, $N_{add}$. Unsurprisingly, reducing the neurogenesis rate leads to weaker memories, but as $N_{add}$ is increased, the memory saturates (*Appendix 3—figure 2C*). Our choice of $N_{add}$ is in the saturated regime. We then looked at the noise parameter $\epsilon$ that represents the activity-independent component of dendritic elaboration (*Appendix 3—figure 2D*). As would be expected, the amount of noise is inversely related to the memory of the network. More significantly, for low levels of noise, the dendritic elaboration was dominated by MC activity, such that GCs predominantly connected to MCs driven by the enrichment even if the spine dynamics were independent of activity. Such specificity in the dendritic network may be unlikely, so we chose a level of noise that allows the memory to decay to zero. This parameter also influences relearning (*Appendix 3—figure 2H*). In the range we explored, relearning remained faster than the initial learning on average, but the degree to which relearning was faster was much larger in trials with less noise. Likewise, the size of the dendritic network also affects relearning (*Appendix 3—figure 2I*). When we doubled the size of this network, GCs did not need the sensory-dependent dendritic elaboration to learn the odors (*Figure 3B*), and thus did not leverage the advantage this mechanism provides unless the amount of noise was low. In this scenario, however, abGCs that had already fully developed their dendrites at the onset of enrichment were responsible for learning, inconsistent with experiments (*Forest et al., 2020*; *Forest et al., 2019*) and the results of *Figure 3C*.

The final parameters we tested were involved in the synaptic plasticity rule. First, we looked at the relative rate of consolidation to deconsolidation. Within the range we tested, we saw no significant change in learning ability (*Appendix 3—figure 2*), suggesting that this parameter does not substantially affect any of the results. Finally, we tested the parameter $k$ in the sliding threshold $\theta_j^{\pm}$. Compared to other parameters shown, small changes in $k$ resulted in more significant changes. If $k$ was too small, GCs did not prune synapses that were not beneficial to processing the odors, leading to non-specific connectivity and poor learning (*Appendix 3—figure 2F*). Alternatively, if $k$ was too large, neurons started to have difficulty consolidating beneficial synapses, also harming learning.

## Acknowledgements

This work was supported by the NSF (DMS-1547394) and NIH (DC015137). BS was supported by a John N Nicholson fellowship.

## Additional information

### Funding

| Funder | Grant reference number | Author |
|---|---|---|
| National Science Foundation | DMS-1547394 | Hermann Riecke |
| National Institute on Deafness and Other Communication Disorders | DC015137 | Hermann Riecke |
| Northwestern University | John N Nicholson fellowship | Bennet Sakelaris |

The funders had no role in study design, data collection and interpretation, or the decision to submit the work for publication.

### Author contributions

Bennet Sakelaris, Conceptualization, Formal analysis, Investigation, Methodology, Software, Visualization, Writing – original draft, Writing – review and editing; Hermann Riecke, Conceptualization, Resources, Software, Formal analysis, Supervision, Funding acquisition, Validation, Investigation, Visualization, Methodology, Writing – original draft, Project administration, Writing – review and editing

### Author ORCIDs

Bennet Sakelaris https://orcid.org/0000-0001-8798-584X
Hermann Riecke https://orcid.org/0000-0002-6070-4742

Reviewer #1 (Public review): https://doi.org/10.7554/eLife.104443.3.sa1
Reviewer #2 (Public review): https://doi.org/10.7554/eLife.104443.3.sa2
Reviewer #3 (Public review): https://doi.org/10.7554/eLife.104443.3.sa3
Author response https://doi.org/10.7554/eLife.104443.3.sa4

## Additional files

### Supplementary files

MDAR checklist

### Data availability

Code for the model is available at https://doi.org/10.21985/n2-mrtc-9v47.

The following dataset was generated:

| Author(s) | Year | Dataset title | Dataset URL | Database and Identifier |
|---|---|---|---|---|
| Riecke H, Sakelaris B | 2025 | Adult Neurogenesis Reconciles Flexibility and Stability of Olfactory Perceptual Memory | https://doi.org/10.21985/n2-mrtc-9v47 | Northwestern University Research and Data Repository, 10.21985/n2-mrtc-9v47 |

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

# Appendix 1

## Discriminability

In addition to using a connectivity-based learning measure, we use an activity-based learning measure to characterize to what extent learning enhances the ability of downstream cortical neurons to discriminate between the odors based on their read-out of the MC activities. Because the MC rate model does not include any fluctuations in activity that would limit discriminability, we assume that the rates represent the mean values of independent Poisson spike trains for which the variance is given by their mean. We assume a linear read-out of the MC activities with the weights chosen optimally and characterize the discriminability of stimuli $A$ and $B$ in terms of the optimal Fisher discriminant $F_{opt}$ **Adams et al., 2019**.

$$F_{opt} = \sum_{i=1}^{N_{MC}} \frac{(M_i^{(A)} - M_i^{(B)})^2}{M_i^{(A)} + M_i^{(B)}}. \tag{A1}$$

Thus, it can be seen that $F_{opt}$ will increase with the addition of MCs, reflecting the fact that even poorly discriminating MCs provide some additional information about the odors.

To verify that our connectivity-based measure of memory aligns with the function of the OB, we calculated the time course of the Fisher discriminant using the data that generated the results in **Figure 2C** (**Figure 2—figure supplement 1B**). Indeed, both measures yield qualitatively similar results, with the fast network learning and forgetting quickly, the slow network learning and forgetting slowly, and the age-dependent network learning quickly and forgetting slowly. Likewise, the neurogenic and non-neurogenic networks performed similarly.

In this study, we focused on the changes in the network connectivity rather than changes in MC activity. We therefore assessed the behavior of the system mostly in terms of the connectivity-based memory. This measure for the memory is agnostic with respect to the odor code, i.e. it does not depend on the type of read-out of the OB activity used by the animal (e.g. rate-based or timing-based **Wilson et al., 2017**; **Bolding and Franks, 2017**).

# Appendix 2

## Comparison with other methods resolving the flexibility-stability dilemma

Previous theoretical work has established a general framework in order to track the memory of an arbitrary stimulus in a stream of random uncorrelated stimuli based only on the properties of the network, without explicitly modeling neuronal activity. It has been used to evaluate models that confront the flexibility-stability dilemma (*Fusi et al., 2005*; *Roxin and Fusi, 2013*; *Benna and Fusi, 2016*; *Fusi, 2021*). In networks with $N$ simple synapses where plasticity occurs on a uniformly fast time scale, the initial memory grows as $\sqrt{N}$ while overall memory capacity grows only logarithmically with $N$ (*Amit and Fusi, 1994*; *Fusi and Abbott, 2007*). Meanwhile, the complex synapses of the cascade model (*Fusi et al., 2005*) and the bidirectional cascade model (*Benna and Fusi, 2016*) as well as the heterogeneity and structure of the partitioned memory system model *Roxin and Fusi, 2013* have been shown to allow the network to achieve far greater capacity. In the case of the cascade model and the partitioned memory system model, memory capacity on the order of $\sqrt{N}$ can be achieved, and in the case of the bidirectional cascade model, memory capacity on the order of $N$ can be achieved, though the latter requires a great degree of complexity in the synapses.

To understand the scaling properties of our model and how they compare with other models, we situated it within this framework. More specifically, we consider a network initially with $N$ binary synapses. At each time step, each synapse is independently presented with a plasticity event, which attempts to flip the synapse depending on the presented stimulus and is accepted with probability $q_i$, the plasticity rate of synapse $i$ (*Appendix 2—figure 1A*). To quantify memory performance, we tracked the signal-to-noise ratio (SNR) of a single arbitrary stimulus previously encoded by the network (*Appendix 2—figure 1B*). We report the flexibility as the SNR immediately after stimulus presentation (the 'initial memory' SNR(0)). The stability we characterize in terms of the time $T$ that it took for the SNR to decay to the value of 1 due to the storage of subsequent memories (the 'memory lifetime'), which can be interpreted as the memory capacity. The presented stimuli are random and uncorrelated. Thus, on average, the initial memory signal $\mu_i(t = 0)$ of a stimulus associated with synapse $i$ is $q_i$ and the total initial memory signal $\mu(0)$ of the network is

$$\mu(0) = \sum_{i=1}^{N} \mu_i(0) = \sum_{i=1}^{N} q_i. \tag{A2}$$

Meanwhile, as this is a system of binomially distributed variables, the variance of the signal can be roughly approximated as $\sqrt{N}$ (*Roxin and Fusi, 2013*), leading to a signal-to-noise ratio $SNR(0)$ of

$$SNR(0) = \frac{1}{\sqrt{N}} \sum_{i=1}^{N} q_i. \tag{A3}$$

As *Roxin and Fusi, 2013* show, the dynamics of the signal to noise ratio of the memory can be described by

$$\frac{dSNR}{dt} = \frac{1}{\sqrt{N}} \sum_{i=1}^{N} \frac{d\mu_i}{dt} \tag{A4}$$

where the $\mu_i(t)$ follow the equations

$$\frac{d\mu_i}{dt} = -q_i \mu_i. \tag{A5}$$

To incorporate the key element of our plasticity model, we extend this model by making the plasticity rates $q_i$ depend on the ages of the cells such that

$$q_i = \begin{cases} q^{fast} & \text{if age of GC } i \text{ is } \leq T_c \text{days} \\ q^{slow} & \text{if age of GC } i \text{ is } > T_c \text{days}, \end{cases} \tag{A6}$$

where $q^{fast} \gg q^{slow}$ and $T_c$ is the duration of the critical period during which the synapse is highly plastic. Following **Fusi et al., 2005**, we choose $q^{fast} \sim \mathcal{O}(1)$ and $q^{slow} \sim \mathcal{O}(N^{-\frac{1}{2}})$. If, at the time of the stimulus presentation, the fraction of synapses on young GCs is $k$, then according to **Equation A3** the *SNR* of that memory is given by

$$SNR(0) = \sqrt{N}(kq^{fast} + (1-k)q^{slow}). \tag{A7}$$

If $kq^{fast} \gg q^{slow}$, the initial memory is controlled by the fast plasticity rate, $SNR(0) \sim \sqrt{N}kq^{fast}$. Indeed, in the rat brain, the total number of GCs is a few million, while roughly 10,000 more are born on each day **Kaplan et al., 1985**. Since the critical period lasts about 14 days, about 140,000 GCs have enhanced plasticity, so $k$ is on the order of 0.1. Thus, assuming $N \sim 10^8$, $kq^{fast} \approx 10^{-1} \gg q^{slow} \approx 10^{-4}$ is a valid assumption.

The memory duration is determined by the time $T$ at which $SNR(t)$ falls below some fixed threshold $\theta_{SNR}$. Solving, we have that for $t \geq T_c$, $SNR(t)$ is given by

$$SNR(t) = \left[ q^{slow}(1-k)\sqrt{N} + q^{fast} \frac{k\sqrt{N}}{T_c} \sum_{j=0}^{T_c} e^{(q^{slow} - q^{fast})j} \right] e^{-q^{slow}t}. \tag{A8}$$

We use this to solve for the memory duration $T$ where $SNR(T) = \theta_{SNR}$. Again, using $q^{slow} \sim \mathcal{O}(\frac{1}{\sqrt{N}})$ and $q^{fast} \sim \mathcal{O}(1)$ we get

$$T = \sqrt{N}\log\left[ \frac{1-k}{\theta_{SNR}} + \frac{k\sqrt{N}}{T_c \theta_{SNR}} \sum_{j=0}^{T_c} e^{(\frac{1}{\sqrt{N}}-1)j} \right]. \tag{A9}$$

For large $N$, memory duration scales approximately as $T \sim \mathcal{O}(\sqrt{N}\log(\sqrt{N}))$, where the leading $\sqrt{N}$ arises from the inverse of $q^{slow}$. This shows that while initial memory is controlled by the fast plasticity of immature synapses, memory duration is controlled by the slow plasticity rate of mature synapses.

We verified these results computationally. First, we compared the memory decay of our model to those of homogeneous models as well as the cascade model and the partitioned memory system model for a network of approximate size to the rat OB (**Appendix 2—figure 1C**). We show our model (red) has a similar initial memory and memory duration as the cascade model (blue) and the partitioned memory system model (green), all of which far outpace the initial memory of the slow-synapse model (gray) and the memory duration of the fast-synapse model (black). Notably, this reiterates that the increased memory capacity provided by neurogenesis is due to the age-dependent properties of adult-born neurons rather than the addition of neurons alone, and that this age dependence can most efficiently utilize the new synapses provided by adult neurogenesis.

Finally, we examine how the initial memory and the memory duration scale with $N$ (**Appendix 2—figure 1D**). We confirm that both the initial memory (**Appendix 2—figure 1E**) as well as the memory lifetime approximately follow $\sqrt{N}$ (**Appendix 2—figure 1F**). Thus, like the cascade model and the partitioned-memory model, our age-dependent model robustly resolves the plasticity-flexibility dilemma, simultaneously achieving the greatest initial memory and memory duration possibly afforded by the homogeneous network with constant plasticity.

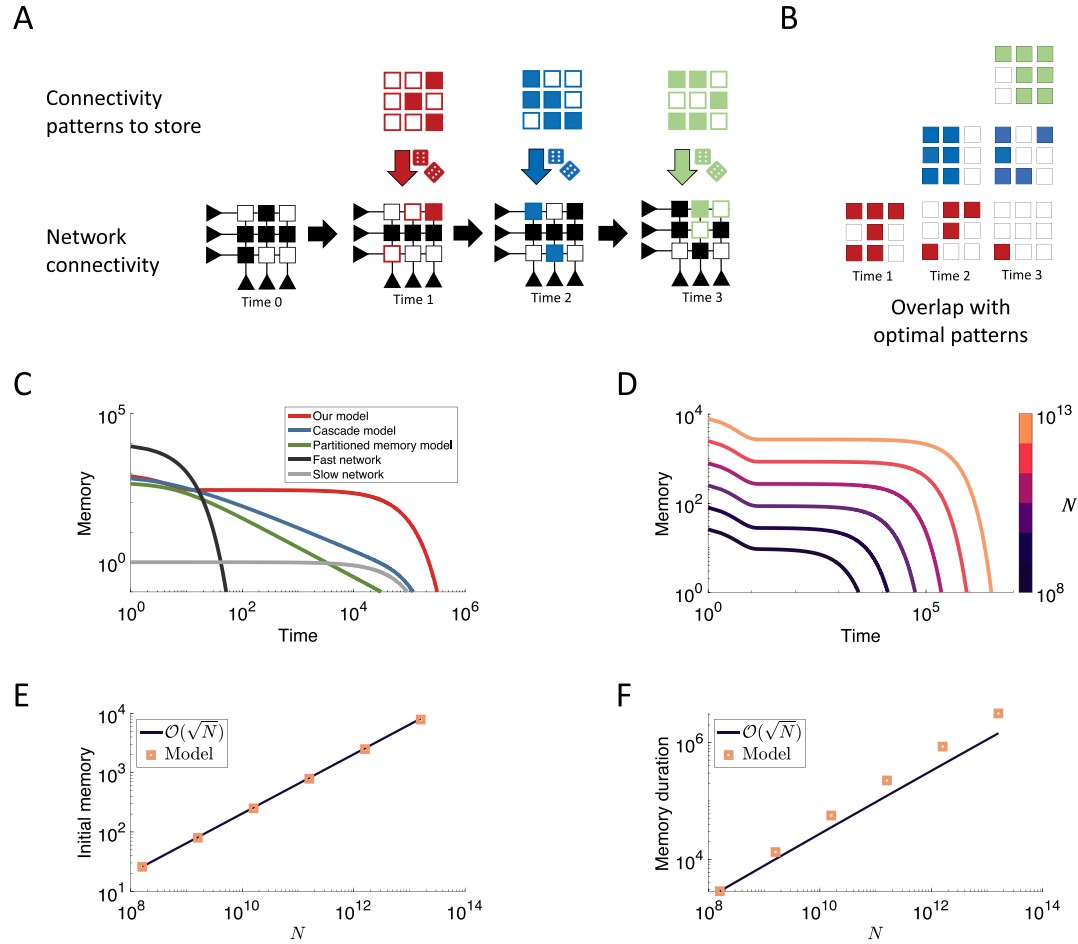

**Appendix 2—figure 1.** Mean-field model. (**A**) We assume there exists an optimal configuration that can process a given stimulus. In this framework, the network directly encodes this configuration stochastically according to the plasticity rate at each synapse, and at each time point, a new stimulus is presented to the network. We track the memory of the network as the degree of overlap between the optimal network for a given stimulus and the current configuration of the network (see Appendix 'Comparison with other methods resolving the flexibility-stability dilemma'). Note that a lack of connection can also represent an overlap. (**B**) Overlap between each stimulus and the current configuration of the network in (**A**). (**C**) Results of the mean-field approximation to the model described in (**A**) with age-dependent synaptic plasticity rates have similar initial memory and memory duration as the cascade model *Fusi et al., 2005*, and the partitioned-memory model *Roxin and Fusi, 2013*. (**D**) Results of the age-dependent model for different values of the number of synapses $N$. (**E**) Initial memory as a function of $N$. (**F**) Memory duration as a function of $N$.

## Appendix 3

### Additional figures

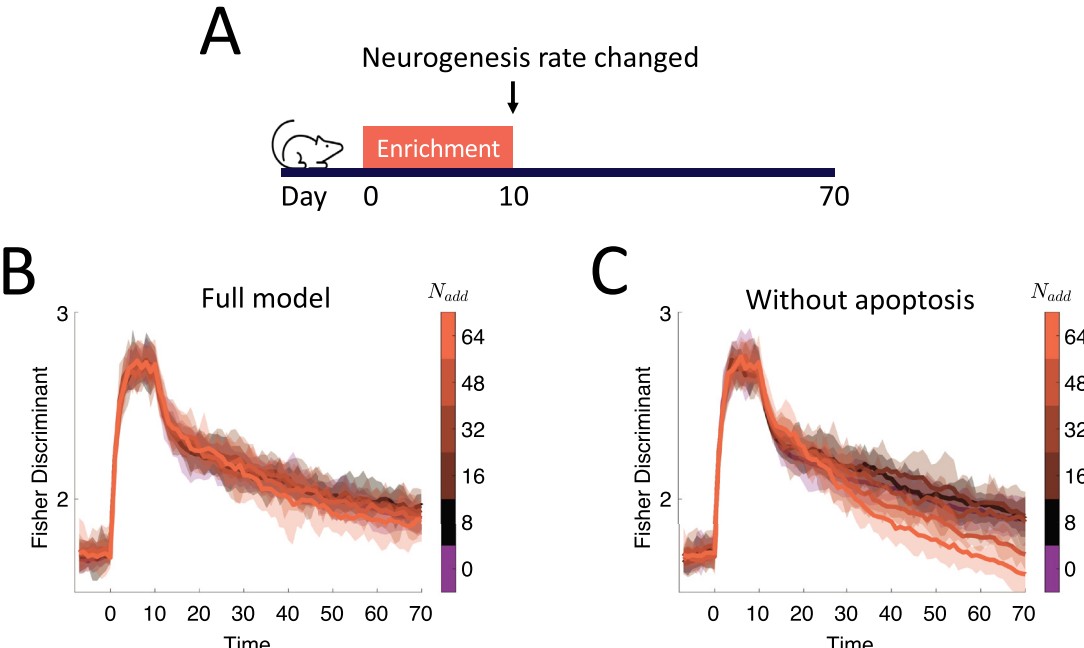

**Appendix 3—figure 1.** Post-learning changes in neurogenesis rate. (**A**) Simulation protocol. Following a 10-day enrichment (using the odors in *Figure 2B*), the neurogenesis rate was permanently changed. (**B, C**) Fisher discriminant between the two similar odors for the full model and the model without apoptosis. The Fisher discriminant was chosen in order to investigate the degree that abGCs interfere with MC activity, which represents the output of the network. The full model can tolerate the addition of vast numbers of new neurons without substantially affecting memory. Without apoptosis, the accumulation of neurons has substantial impact on memory. Lines: mean across eight trials, shaded areas: full range.

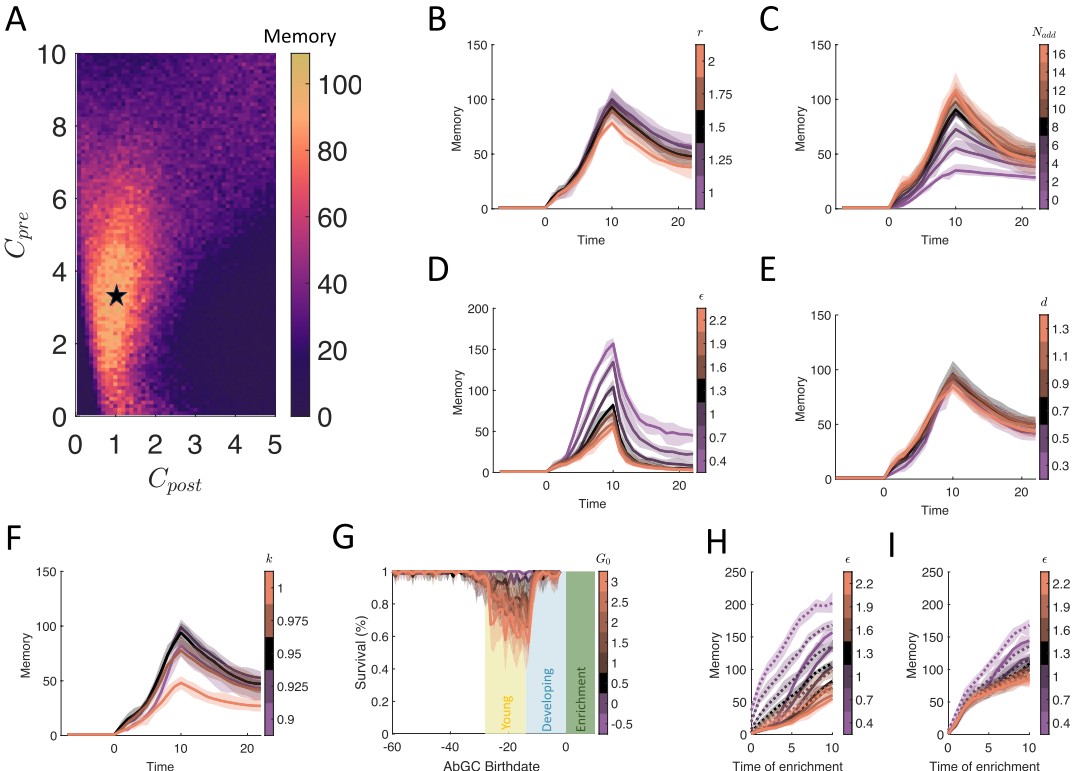

**Appendix 3—figure 2.** Parameter sensitivity. In all plots, the values indicated in black are the parameter values used throughout this study. Lines indicate the mean and shaded areas represent the range over eight trials. (**A**) Final memory following the standard enrichment experiment as a function of $C_{pre}$ and $C_{post}$. (**B–F**) Memory trace over the course of the standard enrichment experiment for different values of $r$ (for abGCs in their critical period), $N_{add}$, $\epsilon$, $d$, and $k$, respectively. Lines: mean across eight trials, shaded areas: full range. In (**D**), $R_0$ was increased to 1 following enrichment to illustrate the final memory value after forgetting. (**G**) GC survival following enrichment for different values of $G_0$ during the critical period of the abGCs (**Figure 4C**). Lines: mean across eight trials, shaded areas: full range. (**H**) Results of the relearning experiment **Figure 3F** for different values of $\epsilon$. Solid: initial learning, dashed: re-learning. Lines: mean across eight trials, shaded areas: full range(**I**) As in (**H**) but for $N_{conn} = 60$ instead of 30.

