## [Editor Report · eLife Assessment]

In this **important** study, the authors use computational modeling to explore how fast learning can be reconciled with the accumulation of stable memories in the olfactory bulb, where adult neurogenesis is prominent. Their model demonstrates that changes in excitability, plasticity, and susceptibility to apoptosis during the maturation of adult-born granule cells can help resolve the flexibility-stability dilemma. These **compelling** results provide a coherent picture of a neurogenesis-dependent learning process that is consistent with diverse experimental observations and may serve as a foundation for further experimental and computational studies.

---

## [Referee Report · Reviewer #1 (Public review)]

Summary:

Sakelaris and Riecke used computational modeling to explore how neurogenesis and sequential integration of new neurons into a network support memory formation and maintenance. They focus on the integration of granule cells in the olfactory bulb, a brain area where adult neurogenesis is prominent. Experimental results published during recent years provide an excellent basis to address the question at hand by biologically constrained models. The study extends previous computational models and provides a coherent picture of how multiple processes may act in concert to enable rapid learning, high stability of memories, and high memory capacity. This computational model generates experimentally testable predictions and is likely to be valuable to understand roles of neurogenesis and related phenomena in memory. One of the key findings is that important features of the memory system depend on transient properties of adult-born granule cells such as enhanced excitability and apoptosis during specific phases the development of individual neurons. The model can explain many experimental observations, and suggests specific functions for different processes (e.g., importance of apoptosis for continual learning). While this model is obviously a massive simplification of the biological system, it conceptualizes diverse experimental observations into a coherent picture, it generates testable predictions for experiments, and it and will likely inspire further modeling and experimental studies.

Strengths:

- The model can explain diverse experimental observations

- The model directly represents the biological network

Weaknesses:

- As many other models of biological networks, this model contains major simplifications.

---

## [Referee Report · Reviewer #2 (Public review)]

Summary:

The authors propose a mechanism to provide flexibility to learn new information while preserving stability in neural networks by combining structural plasticity and synaptic plasticity.

Strengths:

An intriguing idea, well embedded in experimental data.

Authors have done a great job addressing reviewers' concerns

Weaknesses:

None

---

## [Referee Report · Reviewer #3 (Public review)]

The manuscript is focused on local bulbar mechanisms to solve the flexibility-stability dilemma in contrast to long range interactions documented in other systems (hippocampus-cortex). The network performance is assessed in a perceptual learning task: the network is presented with alternating, similar artificial stimuli (defined as enrichment) and the authors assess its ability to discriminate between these stimuli by comparing the mitral cell representations quantified by Fisher discriminant analysis. The authors use enhancement in discriminability between stimuli as function of the degree of specificity of connectivity in the network to quantify the formation of an odor-specific network structure which as such has memory - they quantify memory as the specificity of that connectivity.

The focus on neurogenesis, excitability and synaptic connectivity of abGCs is topical, and the authors systematically built their model, clearly stating their assumptions and setting up the questions and answers. In my opinion, the combination of latent dendritic representations, excitability and apoptosis in an age-dependent manner is interesting and as the authors point out leads to experimentally testable hypotheses.

In the revised manuscript, the authors have systematically addressed my previous concerns. In particular, they now refer to previous work on granule cells-mitral cell interactions more generally, they explain the pros and cons for usage of specificity in connectivity as a proxy for memory capacity, and the biological plausibility of the model.

---

## [Author Response]

The following is the authors’ response to the original reviews

**Reviewer #1:**
(1) Figure 2 and related text: it would be useful to explain more explicitly what is meant by "neurogenic" and "non-neurogenic" models. I presume that the total number of neurons in non-neurogenic models is lower than in neurogenic models because no new neurons are added. It would be useful to plot the number of GCs as a function of timesteps.

We have clarified the distinction between neurogenic and non-neurogenic models in the text (Lines 142-145), explicitly noting that in non-neurogenic models, no new GCs are added, resulting in a lower total neuron count over time. In response to the reviewer’s suggestion, we generated a plot showing the number of GCs over time (see below). Because the neurogenic model exhibits a simple linear increase, we found this plot not especially informative for inclusion in the manuscript. However, we agree with the reviewer’s later comments that similar plots are useful for interpreting specific results, and we have included those where appropriate.

**Author response image 1. sa4fig1:** Number of GCs over time for neurogenic (solid line) and non-neurogenic (dotted line) networks.

(2) Figure 2F, G: memory declines dramatically when the number of GCs at enrichment onset increases beyond an optimum. Why?

We have explained the reasoning more thoroughly in the text (Lines 174-177) and added a new supplemental figure to support this reasoning (Figure S2). As the number of GCs increases, the network becomes overly inhibited and the response of abGCs to the stimuli decreases (Fig S2A). This leads to a smaller population of GCs being able to integrate with the stimulus (Fig S2B) which is expected given the activity-dependent plasticity rule. Moreover, it can be seen in Fig S2C that for networks with increasing size, the GCs that do learn only connect to MCs that are driven strongest by the stimuli until they struggle to connect to any MCs at all.

In principle, a homeostatic mechanism like synaptic scaling could reduce activity to restore balance, but such a mechanism would also likely disrupt existing memories. Alternatively, we suggest activity-dependent apoptosis as a superior homeostatic mechanism because it leads to a stable level of activity without substantially erasing existing memories.

(3) The paragraph describing synaptic connectivity of abGCs (related to Figure 2H) is confusing. What is the directionality of synapses considered here: mitral-to-granule, or granule-to-mitral? The text is opaque here. Connectivity matrix in Figure 2H: who is presynaptic, who is postsynaptic? If I understand correctly, these questions are actually irrelevant because all mitralgranule synapses in the network are reciprocal. This should be pointed out explicitly in the figure legend. Generally: the fact that the network is fully reciprocal (if I understand correctly) is very important but not stated with sufficient emphasis. It should be stated very explicitly in the text that connectivity matrices are fully reciprocal, and an equation clarifying this point should be included in Methods.(6) Connectivity matrix: to what degree was connectivity between mitral and granule cells reciprocal (fraction of connections in either direction that were paired with a connection in the opposite direction between the same cell pair)? Was connectivity shaped by experience (enrichment) reciprocal?(7) Directly related to the above: it would be useful to show the disynaptic connectivity matrix between mitral cells and analyze its symmetry. For the symmetric component, it should then be analyzed what fraction of this can be attributed to the reciprocal synapses, and what fraction is contributed by connectivity via different granule cells. This should then be compared to models with biologically realistic fractions of reciprocal connections. Is the model proposed here consistent with a biologically realistic fraction of reciprocal synapses between mitral-granule cell pairs?

We appreciate these insightful and detailed comments. We agree that the assumption that MC-GC synapses were fully reciprocal was not clearly stated. We now explicitly state this in the main text (lines 90-94, 369-370, Figure 2 caption) and methods (line 561), emphasize its importance. As the reviewer points out, this is a simplifying assumption and does not fully reflect the biology because not all synapses are reciprocal in the true system. We also note that our synaptic plasticity model does not break the reciprocity assumption: all connections added or pruned during learning remain reciprocal. As a result, the disynaptic connectivity matrix (Bottom panel below, MCs sorted by stimulus as shown in the top panel) is always symmetric.

We have now made these statements explicit in the main text and in the methods. Regarding functional consequences of this assumption, earlier work by our group has examined the impact of the degree of reciprocity of MC-GC synapses in a similar OB model (Chow, Wick & Riecke, Plos Comp Bio 2012). The study examined three different changes in reciprocity by (1) redirecting a fraction of the inhibitory connections of each GC to randomly chosen MCs instead of the MCs that drive that GC, (2) allowing heterogeneity in reciprocal weights so that there is no relationship between the strength of the MC -> GC synapse and the GC -> MC synapse, (3) reducing the level of self-inhibition a MC receives from the GCs that it excites. The model was found to be quite robust to each of these manipulations, suggesting that our present model likely remains functionally relevant even if biological reciprocity is partial. We reference this work now in the discussion, lines 490-492.

**Author response image 2. sa4fig2:** Disynaptic connectivity. Top: MC activity in response to the two stimuli, sorted by MC selectivity. Bottom: Disynaptic connectivity matrix (diagonal subtracted).

(4) How were mitral cells sorted in Figure 2H? This needs to be explained.(5) Directly related to the point above: the text mentions that synaptic connectivity between GCs of the "learning cluster" and mitral cells (which direction?) is increased for mitral cells responding by enrichment odors, but this is not shown in the figure. This statement suggests that mitral cells sorted to the bottom of the y-axis respond more strongly to enrichment odors, but the information is not given directly. Please provide more information to back up your statements.

Indeed as the reviewer inferred, MCs in Figure 2H were sorted so that those that receive the strongest stimulation from the odor were at the bottom of the y-axis. We have clarified this in the Figure 2 caption and added a subplot to Figure 2H showing the average MC input to make this more explicit.

(8) Apoptosis (Figure 4 and related text): paragraph 231ff is somewhat difficult to comprehend because the "number" of enrichments should really be the "frequency" of enrichments. In Figure 4, it is not mentioned explicitly that each enrichment is with different random new odors.

We agree that the term “number” of enrichments was imprecise and have revised the text to refer instead to the frequency of enrichment events (Lines 255-267). We also clarified that in Figure 4, each enrichment corresponds to a different set of randomly sampled odors, and we now state this explicitly in both the Figure 4 legend and main text (Lines 260-261).

(9) Apoptosis: apoptosis improves memory but the underlying reason remains opaque. A simple prediction of the data in Figure 4D and 4E is that the number of GCs in 4E. It would be helpful to show this. Furthermore, an obvious question that arises is whether a higher frequency of enrichments improves memories because the total number of granule cells is kept low, or because granule cells are removed specifically based on their activity (or both). This could be addressed easily by artificially removing a random subset of granule cells in a simulation such as 4E to match granule cell numbers to the case in 4D.

Apoptosis improves learning is because it reduces the total inhibition in the network by removing GCs and thus prevents deficits in learning that occur in Fig. 2G as GCs accumulate in the network. As the reviewer inferred, the number of GCs in Figure 4D is lower than in 4E and this is now clarified in the text. This difference was shown implicitly in Supplementary Figure S4D (previously S3D), but we now explicitly reference this plot to support this point as well (Line 266).

As the reviewer notes, there is a question in whether increased enrichment frequency improves memory because it limits the total number of GCs, or because apoptosis selectively removes GCs based on their activity, or both. Our model supports both mechanisms. Importantly, simply reducing GC numbers through random deletion will degrade existing memories: random removal erodes memory representations encoded by those GCs. In contrast, our age and activity dependent apoptosis rule targets a specific cohort of adult-born GCs. This selective removal minimizes damage to existing memories encoded by GCs outside of this cohort while keeping GC numbers within a regime that supports robust learning (as shown in Figure 2G).

However, we note that if enrichment frequency becomes too high, even recent memories can be lost due to premature pruning of GCs that have not yet stabilized their synaptic connections. This tradeoff has been shown experimentally (Forest et al., Nat Comm 2019) which we reproduce in our model (Figure S4).

(10) Text related to Figure 5: "Learning flexibility...approached a steady state when the growth of the network started to saturate". Please show the growth (better: size) of the network (total number of GCs) for these simulations (and other panels in Figure 5). It would also be useful to show the total number of GCs in other figures (e.g. Figure 4; see above).

We have now added a supplementary figure (Figure S6) that shows the total number of GCs over time for the simulations presented. This confirms that the network size approaches a steady state around the same time that learning flexibility begins to plateau, as noted in the original text (now line 275), and highlights the large number of GCs without apoptosis as well as the slightly reduced number of GCs in the permanent encoding model (line 312).

(11) As much as I appreciate the comprehensive discussion of the results in a broader context, I feel that the discussion can be somewhat shortened. The section on lateral inhibition is not fully valid given that synaptic connectivity is reciprocal. I also feel that much of the final section (Model assumptions and outlook) can be dropped (except for the last paragraph), not because anything is irrelevant, but because these points have been made, onen repeatedly, in the text above.

We agree that the discussion could be streamlined and have revised the manuscript accordingly. Specifically, we have shortened the section on lateral inhibition and clarified that the OB relies predominantly on reciprocal connectivity (Line 370). We also agree that parts of the final section were repetitive and have removed these. However, to address comments by Reviewer 3, we also expanded on some of the model assumptions. We thank the reviewer for helping us improve the clarity and focus of the manuscript.

(12) Figure 5: bolding every 5th curve is confusing.

We have adjusted our figure accordingly.

(13) "...we biased the dendritic field...": it would be helpful to explain the idea of a "dendritic field" in a bit more detail prior to this sentence.

We have now noted that GC’s "dendritic field" refers to the subset of MCs with which it is capable of forming synaptic connections when we initially describe the model (Line 97).

**Reviewer #3:**
(1) The authors find that a network with age-dependent synaptic plasticity outperforms one with constant age-independent plasticity and that having more GC per se is not sufficient to explain this effect. In addition, having an initial higher excitability of GCs leads to increased performance. To what degree the increased excitability of abGCs is conceptually necessarily independent of them having higher synaptic plasticity rates / fast synapses?

We thank the reviewer for this question, as the difference between excitability and plasticity rate in memory formation is something we intended to highlight in this study. We have updated the (Lines 157-198) to clarify this.

At the cellular level, a neuron's excitability and its rate of synaptic plasticity are mechanistically distinct: excitability is governed by factors such as ion channel expression or membrane resistance, whereas plasticity rates are influenced by molecular pathways involved in synapse and dendritic spine formation and remodeling. While these are independent properties, they are functionally coupled: most synaptic plasticity rules are activity-dependent, so greater excitability can increase the likelihood of plasticity being induced but does not itself guarantee learning.

Our model reflects this distinction. Increased excitability biases which neurons become activated and thus eligible to undergo plasticity, but actual learning still depends on the plasticity rate itself. This can be seen by comparing the model constant plasticity and excitability (solid blue and green curves in Figure 2C) to the model with only transient excitability (solid blue and green lines in Figure 2E). In both cases, the strength and duration of the memory remain limited by the plasticity rate. We note additionally that, in this network, neurons compete to learn new stimuli: as GCs start to learn, they suppress MC activity through recurrent inhibition which suppresses learning in other GCs who otherwise would have been in position to learn the odor. As a result there is not a significant increase in the overall number of neurons recruited to learn (Figure 2J). In a different network architecture, such as a feedforward network, we would not expect this to be the case; greater excitability in a population of neurons would likely increase the memory by increasing the number of neurons recruited to learn. Transiently enhanced excitability biases which neurons join the memory engram (Figure 2J), but the extent and rate of learning still depend on the plasticity rates themselves. We did note in the original text (now lines 284-286) that this bias in recruitment subtly increases memory stability, but the extent is not great. In principle, a model can be engineered to rely on transiently increased excitability to encode memories in orthogonal subpopulations of neurons and that this could resolve the flexibility-stability dilemma. However, in that case, the number of memories that can be stored within a short time would be bounded by the size of this subpopulation such that even if a large number of odors are presented, mature GCs cannot become part of the engram and the network would likely fail to learn the stimuli. However, when this was tested experimentally (Forest et al. Cereb Cor. 2020), it was found that mature GCs participated in the engram when the number of odors was sufficiently high. Our results are consistent with these experiments: for complex odor environments, neonatal GCs, which are mature during odor exposure, and abGCs both participate in the engrams.

**Author response image 3. sa4fig3:** Simulating learning in more complex odor environments. Top: enrichment consisted of three odor pairs presented sequentially in a random order. Bottom: enrichment consisted of five odor pairs. Left: discriminability of the odor pairs over time. Middle: connectivity between MCs (sorted by odor selectivity) and GCs (sorted by age). In both cases AbGCs develop a clear connectivity structure. In more complex environments neonatal GCs also start to develop a clear connectivity structure. Right: combined engram membership across all stimuli by GC age.

In sum, transiently increased excitability alone will not make learning any faster, so a fast learning system must have a high plasticity rate. If this plasticity rate stays high, then memories stored in these neurons, even if no longer highly excitable, will be vulnerable as the neurons can still be driven above their plasticity threshold by moderately interfering stimuli and will thus be quickly forgotten. Conversely, if the reviewer is wondering if a greater increase in the plasticity rate of new neurons can compensate for a lack of excitability, this is not the case: if a newborn neuron is not sufficiently driven by the stimulus it will not learn regardless of how high its plasticity rate is.

(2) The authors do not mention previous theoretical work on the specificity of mitral to granule cell interactions from several groups (Koulakov & Rinberg - Neuron, 2011; Gilra & Bhalla, PLoSOne, 2015; Grabska-Bawinska...Mainen, Pouget, Latham, Nat. Neurosci. 2017; Tootoonian, Schaefer, Latham, PLoS Comput. Biol., 2022), nor work on the relevance of top-down feedback from the olfactory cortex on the abGC during odor discrimination tasks (Wu & Komiyama, Sci. Adv. 2020), or of top-down regulation from the olfactory cortex on regulating the activity of the mitral/tuned cells in task engaged mice (Lindeman et al., PLoS Comput. Biol., 2024), or in naïve mice that encounter odorants (in the absence of specific context; Boyd, et al., Cell Rep, 2015; Otazu et al., Neuron 2015, Chae et al., Neuron, 2022). In particular, the presence of rich topdown control of granule cell activity (including of abGCs) puts into question the plausibility of one of the opening statements of the authors with respect to relying solely on local circuit mechanisms to solve the flexibility-stability dilemma. I think the discussion of this work is important in order to put into context the idea of specific interactions between the abGCs and the mitral cells.

We thank the reviewer for these detailed and thorough comments, and whole-heartedly agree that it is important to discuss the listed studies in order to contextualize our work through the broader lens of how information is processed in the OB. We have expanded our discussion to further acknowledge and integrate insight from previous theoretical and experimental work cited by the reviewer. (Lines 361-366, 493-550)

Regarding the importance of top-down feedback, we of course recognize that in practice cortical inputs play a critical role in abGC survival and synaptic integration. However, its nature is not quite clear and is likely variable across behavioral seungs. In the paradigm that we study in the manuscript, there is likely no key reward value or contextual signal that is relayed to the OB. One plausible interpretation is that in this task, cortical feedback provides a random, variable baseline excitatory drive to GCs. This would likely be consistent with many of the listed studies, e.g.

(1) Glomerular layer targeting of feedback would be explicitly unrelated to glomerular odor specificity, as in Boyd et al.

(2) GC activity would decrease if these cortical inputs were silenced, resulting in stronger MC responses as in Otazu et al., Chae et al.

(3) Silencing PCx during learning would prevent GCs from reaching activity-dependent plasticity thresholds, resulting in decreased spine density as in Wu & Komiyama.

Likewise activating PCx would lead to increased spine density.

In this interpretation, the effect of top-down input could be captured implicitly by adjusting model parameters such as activity or plasticity thresholds. For the purposes of our study, we opted to neglect these inputs in favor of model simplicity.

Critically, even if top-down inputs play a substantially larger role, by perhaps even going as far as providing signals to abGCs to modulate their development, the core solution to the flexibility-stability dilemma that we describe stays local: we predict that the memory persists in the same network in which it was formed.

(3) To what the degree of specific connectivity reflects a specific stimulus configuration, and is a good proxy for determining the stimulus discriminability and memory capacity in terms of temporal activity patterns (difference in latency/phase with respect to the respiration cycle, etc.) which may account to a substantial fraction of ability to discriminate between stimuli? The authors mention in the discussion that this is, indeed, an upper bound and specific connectivity is necessary for different temporal activity patterns, but a further expansion on this topic would help in understanding the limitations of the model.

We thank the reviewer for raising this important point. Indeed, there have been several recent experimental studies indicating that much of the information needed for olfactory discrimination is encoded in the temporal activity patterns of mitral and tuned cells. Our model does not explicitly simulate these dynamics. It was for this reason that we defined memory in terms of the learned structure of the network rather than by firing rate activity. This is motivated by the idea that learned patterns of connectivity constrain the space of neural activity the network can support, and thus shape stimulus responses. We now make this limitation more explicit in the discussion and clarify that the specific MC–GC connectivity we analyze should be seen as a structural substrate that constrains the possible temporal transformations the network could support (Lines 492-506).

(4) Reward or reward prediction error signals are not considered in the model. They however are ubiquitous in nature and likely to be encountered and shape the connectivity and activity patterns of the abGC-mitral cell network. Including a discussion of how the model may be adjusted to incorporate reward/error signals would strengthen the manuscript.

We appreciate the reviewer’s suggestion and agree that reward and reward prediction error signals are critical components of many learning paradigms. We deliberately chose not to model associative learning, reward signals or top-down neuromodulation in this work. Our goal is to investigate the role of adult neurogenesis in a regime where its contribution has been shown to be experimentally necessary. Specifically, we focused on an unsupervised perceptual learning paradigm where adult neurogenesis is required for successful odor discrimination (Moreno et al. PNAS, 2008). In contrast, when the same odors are used in a rewarded learning paradigm, performance remains intact even when adult neurogenesis is ablated (Imayoshi et al., Nat. Neuro., 2008). This dissociation suggests that neurogenesis is dispensable in contexts where reward can guide learning. As such, we argue that isolating the contribution of local circuit dynamics in an unsupervised setting is critical to understanding what neurogenesis is uniquely enabling, especially given the evolutionary cost of maintaining it.

We agree that extending this work to incorporate reward-driven plasticity or neuromodulatory influences would be a valuable direction for future research. In particular, it could help clarify how different learning paradigms engage distinct abGC cohorts (e.g., Mandairon et al., eLife 2018; Wu & Komiyama, Sci. Adv. 2020), and how task structure shapes memory allocation and engram composition. We have incorporated this into the discussion regarding extending our model to include top down feedback (lines 539-553).

Specific comments(1) Lines 84-86; 507-509; Eq(3): Sensory input is defined by a basal parameter of MCs spontaneous activity (Sspontaneus) and the odor stimuli input (Siodor) but is not clear from the main text or methods how sensory inputs (glomerular patterns) were modeled

We now clarify in the Methods section "Stimulus model" how the sensory inputs were modeled. Specifically, odor-evoked inputs to mitral cells (Siodor) were generated either as Gaussian profiles across the mitral cell population (Figs. 2,3) or as sparser random patterns (Figs. 4,5). In Figures 2 and 3, the denser Gaussian stimuli require more GCs to learn the odors, aiding in visualization of the connectivity matrix (Figure 2H) and abGC recruitment plots (Figure 2I,J; Figure 3C,E). However, real olfactory stimuli activate a sparse set of MCs, so in Figures 4 and 5 where we address learning of many stimuli, we utilize sparser, binary, stimuli delivered to only 10% of MCs, in range of experimental data (Wachowiak and Cohen, Neuron, 2001). The fact that the stimuli are binary, however, is not realistic and leads to denser representations. This leads to a worst-case scenario for the model as denser memory representations are easier to overwrite. These points has been added explicitly to the Methods section "Stimulus model" to improve clarity.

(2) Lines 118-122: The used perceptual learning task explanation is done only in the context of the discriminability of similar artificial stimuli using the Fisher discriminant and "Memory" metric. A detailed description of the logic of the perceptual learning task methods and objective, taking into account Comment 1, would help to better understand the model.

We thank the reviewer for pointing out had not adequately described the task and have updated the main text (lines 125-132) and included a new methods section "Perceptual learning task" to describe it more explicitly. The experiments that inspired the simulation followed an ecological model of discrimination learning (Moreno et al. PNAS 2009): For one hour a day over a ten day "enrichment period", two tea balls containing similar but distinct odors were suspended from the lid of each mouse's home cage. The mice engaged with the stimuli under self-directed conditions, therefore learning through natural experience. As a result the mice use olfactory information to discriminate between the similar stimuli, a skill potentially relevant for navigation or social behaviors.

In our simulations, we model these experiments as follows. During the enrichment period, the model is stimulated with a randomly selected stimulus chosen from a set of two similar stimuli, corresponding to a mouse choosing to sniff one of the tea balls. During enrichment, in between these bouts of "sniffing", the model only receives spontaneous activity, reflecting the temporal sparsity of sensory input even over the enrichment period. Outside of enrichment, the model again receives only spontaneous input.

(3) Rapid re-learning of forgotten odor pair is enabled by sensory-dependent dendritic elaboration of neurons that initially encoded the odors and the observed re-learning would occur even if neurogenesis was blocked following the first enrichment and even though the initial learning did require neurogenesis. When this would ever occur in nature? The re-learning of an odor period? Why is this highlighted in the study?

We believe that this sort of learning is certainly relevant in nature. To clarify: by “learning,” we do not refer to the memory of an entire “odor period”, but simply an altered mapping of specific stimuli. Therefore, forgeung could occur if these specific stimuli are absent from the environment for a period of time, and re-learning would occur when these stimuli are re-encountered. Natural odor environments are highly dynamic, as environmental conditions and social contexts change over time. The odors an animal encounters also depend strongly on its own behavior; as it explores different environments, it may be exposed to particular odors intermittently: it could encounter them in one location, then not return to that location for some time before returning again.

Such natural variability in odor exposure makes the ability to forget and re-learn especially valuable, allowing the animal to prioritize relevant information while maintaining flexibility. To this end, we show in Figure 5G that the synaptic forgetting of odors is beneficial to the performance of the model because it reduces interference in the network. Therefore we highlight that re-learning enabled by adult neurogenesis is a highly efficient strategy for memory storage and retrieval, which is why he emphasize it in this study.

(4) Figure 2A: I understand that the ages shown at the bottom of the colored boxes represent the GC age. If so, find a better way to express that to avoid confusing 'GC ages' from the days shown in the perceptual learning task description (Figure 2B).

We have updated the text in the figure to disambiguate the two and refer to the “days” shown in the perceptual learning task description now as “time relative to enrichment”

(5) Figure 2B: Clarify how the two-dimensional arrays are arranged to represent the patterns shown. Does each point of the array represent one neuron? If so, are these neurons re-arranged to help the readers visually differentiate patterns A and B? Are the patterns of activity of MCs in the model spatially and temporally sparse as observed in experimental work?

In Figure 2B, each point in the two-dimensional array represents the activity of a single mitral cell. The layout is purely for visualization—neurons are re-arranged to make the differences between odor patterns A and B visually apparent. This ordering does not reflect anatomical position or model architecture. We revised the Figure 2 caption to say this explicitly.

Regarding spatial sparseness, as we mentioned in the response to the reviewer’s comment (1), the activity of mitral cells in response to odors is spatially sparse in the model. Regarding temporal sparseness, while the model is not spiking and does not include temporal dynamics within the timescale of the breath, however, odor input is delivered in discrete, odorspecific epochs interleaved with periods of no input, which leads to temporally structured activity patterns. This information has been made explicit in the new methods sections "Stimulus model" and "Perceptual learning task"

(6) Figure 3C and Line 189: potential confusion between the color code mentioned in the legend for the enrichment and developing periods.

It appeared to be a confusion in the text and has been corrected (Lines 212-213).

(7) Figure 5F: For clarity, this would benefit from replacing the bold line with areas in the plot to depict the enrichment periods.

We agree that replacing the bolded line segments with shaded areas is more clear and have updated the figure accordingly, and appreciate the reviewer's suggestion to clarify the figure.

(8) Lines 380, 416: Potential role of cortical feedback and or neuromodulation depending on behavioral relevance or permanent exposure? Later mentioned in Lines 467 - 474.

We have updated the text to acknowledge the role of potential cortical feedback and neuromodulation, now in lines 403-407.